# Lymphocyte innateness defined by transcriptional states reflects a balance between proliferation and effector functions

Maria Gutierrez-Arcelus [1,2,3,4], Nikola Teslovich [1,2,3], Alex R. Mola[3], Rafael B. Polidoro[3], Aparna Nathan[1,2,3,4], Hyun Kim [1,2,3], Susan Hannes[1,2,3,4], Kamil Slowikowski [1,2,3,4], Gerald F.M. Watts[3], Ilya Korsunsky [1,2,3,4], Michael B. Brenner[3], Soumya Raychaudhuri [1,2,3,4,5] & Patrick J. Brennan[3]

How innate T cells (ITC), including invariant natural killer T (iNKT) cells, mucosal-associated invariant T (MAIT) cells, and γδ T cells, maintain a poised effector state has been unclear. Here we address this question using low-input and single-cell RNA-seq of human lymphocyte populations. Unbiased transcriptomic analyses uncover a continuous 'innateness gradient', with adaptive T cells at one end, followed by MAIT, iNKT, γδ T and natural killer cells at the other end. Single-cell RNA-seq reveals four broad states of innateness, and heterogeneity within canonical innate and adaptive populations. Transcriptional and functional data show that innateness is characterized by pre-formed mRNA encoding effector functions, but impaired proliferation marked by decreased baseline expression of ribosomal genes. Together, our data shed new light on the poised state of ITC, in which innateness is defined by a transcriptionally-orchestrated trade-off between rapid cell growth and rapid effector function.

[1] Department of Medicine, Division of Genetics, Brigham and Women's Hospital, Harvard Medical School, Boston, MA, USA 02115. [2] Program in Medical and Population Genetics, Broad Institute, Cambridge, MA 02142, USA. [3] Department of Medicine, Division of Rheumatology, Immunology and Allergy, Brigham and Women's Hospital, Harvard Medical School, Boston, MA 02115, USA. [4] Center for Data Sciences, Brigham and Women's Hospital, Harvard Medical School, Boston, MA 02115, USA. [5] Faculty of Medical and Human Sciences, University of Manchester, Manchester M13 9PL, UK. Correspondence and requests for materials should be addressed to S.R. (email: soumya@broadinstitute.org) or to P.J.B. (email: pbrennan3@bwh.harvard.edu)

Within the spectrum of immune defense, "innate" and "adaptive" refer to pre-existing and learned responses, respectively. Mechanistically, innate immunity is largely ascribed to 'hardwired,' germline-encoded immune responses, while adaptive immunity derives from recombination and mutation of germline DNA to generate specific receptors that recognize pathogen-derived molecules, such as occurs in T and B cell receptors. However, the paradigm that somatic recombination leads only to adaptive immunity is incorrect. Over the past 15 years, T-cell populations have been identified with T-cell antigen receptors (TCRs) that are conserved between individuals. Many of these effector-capable T-cell populations are established in the absence of pathogen encounter. Examples of such T-cell populations include invariant natural killer T (iNKT) cells, mucosal-associated invariant T (MAIT) cells, γδ T cells, and other populations for which we have a more limited understanding[1]. These "donor unrestricted" T-cell populations have been estimated to account for as much as 10–20% of human T cells[2], and have critical roles in host defense and other immune processes. We and others now refer to these cells as innate T cells (ITC).

ITC develop from the same thymic progenitor cells as adaptive T cells, and each of these populations is thought to develop independently. However, ITC populations share several important features that distinguish them from adaptive cells. First, they do not recognize peptides presented by MHC class I and class II. iNKT cells recognize lipids presented by a non-MHC-encoded molecule named CD1d[3]. MAIT cells recognize small molecules, including bacterial vitamin B-like metabolites presented by another non-MHC-encoded molecule, MR1[4]. It is not known whether specific antigen-presenting elements drive the development or activation of γδ T cells. One major γδ T-cell population bearing Vγ2-Vδ9 TCRs is activated by self- and foreign phosphoantigens in conjunction with a transmembrane butyrophilin-family receptor, BTN3A1[5,6]. The antigens recognized by other human γδ T-cell populations are not clear, although a subset of these cells recognizes lipids presented by CD1 family proteins[7]. A second shared feature of ITC is that their responses during inflammation and infection exhibit innate characteristics, such as rapid activation kinetics without prior pathogen exposure, and the capacity for antigen receptor-independent activation. Inflammatory cytokines such as IL-12, IL-18, and type I interferons can activate ITC even in the absence of concordant signaling through their TCRs, and such TCR-independent responses have been reported in iNKT cells[8], MAIT cells[9], and γδ T cells[10].

Given the similar functions reported among different ITC populations, we hypothesize that shared effector capabilities may be driven by common transcriptional programs. Here, using low-input RNA-seq and single-cell RNA-seq, we transcriptionally define the basis of innateness in human ITC by studying them as a group, focusing on their common features rather than what defines each population individually. Using unbiased methods to determine global interpopulation relationships, we reveal as a primary feature an "innateness gradient" with adaptive cells on one end and natural killer (NK) cells on the other, in which ITC populations cluster between the prototypical adaptive and innate cells. Interestingly, we observe a decreased transcription of cellular translational machinery and a decreased capacity for proliferation within innate cell populations. Innate cells rather prioritize transcription of genes encoding for effector functions, including cytokine production, chemokine production, cytotoxicity, and reactive oxygen metabolism. Thus, growth potential and rapid effector function are hallmarks of adaptive and innate cells, respectively.

## Results

**Human ITC immunophenotyping.** To characterize the abundance and variability of ITC in humans, we quantified four major populations of ITC from 101 healthy individuals aged 20–58 years by flow cytometry, directly from peripheral blood mononuclear cells (PBMCs) in the resting state. We assessed the frequencies of iNKT cells, MAIT cells, and the two most abundant peripheral γδ T-cell groups, those expressing a Vδ2 TCR chain (Vδ2) and those expressing a Vδ1 TCR chain (Vδ1). MAIT cells contributed from 0.1 to 15% of T cells (mean 2.4%), iNKT cells from undetectable to 1.1% (mean 0.09%), Vδ1 cells 0.25–6.2% (mean 1.25%), and Vδ2 from 0.08 to 22% (mean 4.7%). The sum of these four cell types accounted for 0.9–25.7% of an individual subject's T cells (mean 8.4%) (Fig. 1a, Supplementary Data 1). Vδ2 cells were more abundant than Vδ1 in 82% of subjects, with the ratio of these two cell types ranging from 0.2 to 67.8 (mean 8.5). Age negatively associated with the total percentage of ITC ($P = 1.4e{-}05$, Pearson correlation, $t$ test). MAIT ($r = -0.42$, $P = 9.9e{-}06$, Pearson correlation, $t$ test) and Vδ2 ($r = -0.43$, $P = 4.7e{-}06$, Pearson correlation, $t$ test) populations drove this association (Supplementary Figure 1a, b), even after accounting for the abundances of other cell types ($P = 5.9e{-}04$, $P = 1.2e{-}04$, respectively, linear regression, $t$ test), which is consistent with previous findings[11,12]. We observed covariance between the frequencies of MAIT and iNKT cells ($P = 0.02$, Spearman correlation, $t$ test), corrected for the other cell types and age (Supplementary Figure 1c, d). We observed no significant associations between ITC percentage and gender ($P = 0.12$, linear regression, $t$ test), body mass index ($P = 0.31$, linear regression, $t$ test), or smoking status ($P = 0.04$, individual cell types $P > 0.2$, linear regression, $t$ test) after accounting for age. Together, these results show that human ITC contribute to a substantial portion

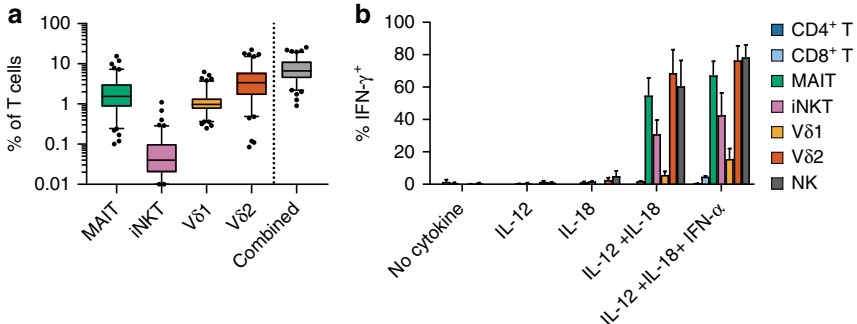

**Fig. 1** ITC immunophenotyping. **a** ITC were quantified in 101 healthy donors by flow cytometry. The "Combined" group represents the sum of iNKT, MAIT, Vδ1, and Vδ2 T cells. For boxplots, 5–95 percentile and outliers are shown. **b** Intracellular staining for IFN-γ production following cytokine stimulation without TCR activation, $N = 3$ independent donors, error is s.e.m.

of the peripheral T-cell repertoire, are variable between individuals, and decrease with age.

**ITC populations rapidly release cytokines**. We next tested innate T-cell populations for two functional hallmarks of innate effectors, rapid cytokine production and TCR-independent activation. To assess rapid cytokine production potential, we activated healthy donor PBMCs with phorbol 12-myristate 13-acetate (PMA) and ionomycin for 4 h, followed by intracellular staining for interferon-γ (IFN-γ) production. Between 35 and 85% of MAIT, iNKT, Vδ1, and Vδ2 T cells produced IFN-γ under these conditions, while a smaller percentage of adaptive CD4$^+$ T and CD8$^+$ T cells produced this cytokine. While a significantly higher proportion of ITC produced IFN-γ compared with adaptive T cells, the difference in mean fluorescence intensity (MFI) did not differ significantly, reflecting both high IFN-γ production among a subset of CD8$^+$ T cells and heterogeneity in the level of IFN-γ production by ITC following activation with PMA and ionomycin (Supplementary Figure 2a–c). To test the relative capacity of these cell types to respond to inflammatory cytokines alone, a hallmark of innate cells, we activated PBMCs with IL-12 + IL-18 or IL-12 + IL-18 + IFN-α for 16 h, and assessed IFN-γ production during the final 4 h of stimulation. Intracellular cytokine staining measured by either percent positive or MFI showed that iNKT, MAIT, Vδ2, and NK cells substantially produced IFN-γ under these conditions, while only a tiny portion of adaptive cells responded (Fig. 1b, Supplementary Figure 2d–f). Taken together, these studies show that ITC populations rapidly produce IFN-γ, and can do so in response to inflammatory cytokines even in the absence of TCR signals. Notably, we observed the latter activation mechanism almost exclusively in ITC populations.

**RNA-seq reveals a continuous innateness gradient**. To better understand the biological properties of human ITC on a genome-wide scale, we profiled their transcriptomes with RNA-seq. Low-input RNA-seq profiling using 1000 cells per sample enabled high-depth sequencing of even relatively rare human lymphocyte populations. From six healthy individuals, we sorted in duplicate four subsets of ITC: iNKT, MAIT (defined as MR1–5-OP-RU tetramer$^+$), Vδ1, and Vδ2 cells (Supplementary Table 1). From the same individuals, we also sorted CD4$^+$ and CD8$^+$ T cells as comparator-adaptive T cells and NK cells as comparator-innate cells (Supplementary Figure 3). Using SmartSeq2 to create poly (A)-based libraries, we generated 25-base-pair, paired-end libraries sequenced at a depth of 4–12 million read pairs (Supplementary Figure 4, Supplementary Data 2). After sequence mapping, we calculated tpm (transcripts per million) values for each gene. We considered 19,931 genes as expressed (tpm > 3 in ≥ 10 samples), including 12,730 protein-coding, 183 T-cell receptor genes, 3261 long noncoding RNA (lncRNA), and other lowly expressed genes (e.g., pseudogenes, Supplementary Figure 4c).

Principal component analysis (PCA) identified the major axes of variation in gene expression (Fig. 2a). The first principal component separated the subsets by a continuous "innateness gradient" with CD4$^+$ and CD8$^+$ T cells on one end, and NK cells on the other end. Ordered from adaptive to innate along the first principal component, MAIT, NKT, Vδ1, and Vδ2 clustered in-between the adaptive cells and NK cells (Fig. 2b). We then identified genes associated with the rank order of each lymphocyte population in the innateness gradient (CD4$^+$ T = 1, CD8$^+$ T = 2, MAIT = 3, iNKT = 4, Vδ1 = 5, Vδ2 = 6, and NK = 7), using linear mixed models. This analysis revealed 1884 genes significantly associated with the innateness gradient (P < 2.5e–06, Bonferroni threshold, likelihood ratio test, see the

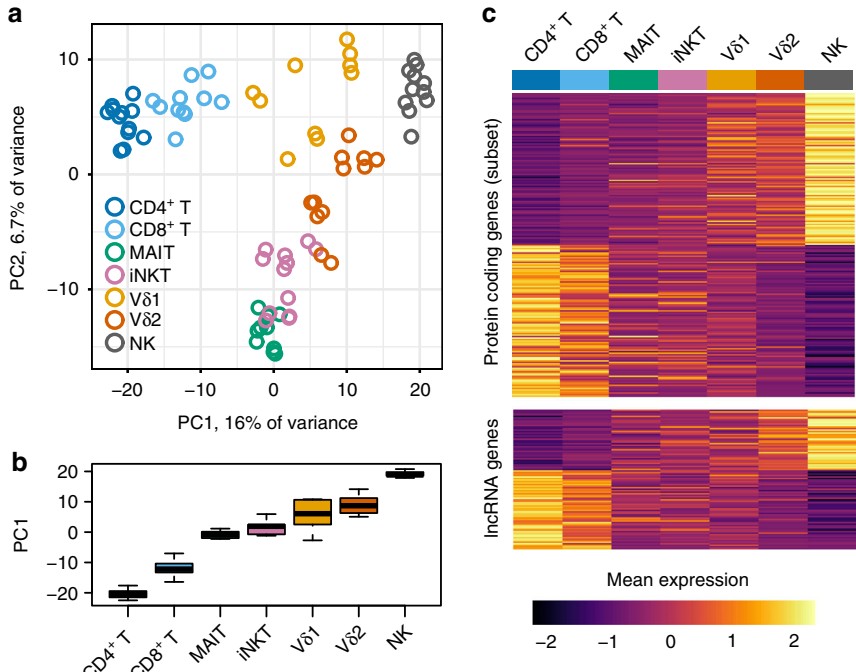

**Fig. 2** Transcriptomic profiling of ITC reveals a continuous innateness gradient. **a** PCA performed on the top 1022 most variable (s.d. > 1.4) expressed genes. Plotted are scores for PC1 and PC2. **b** Distribution of PC1 scores by cell type. **c** Heatplot of mean expression by cell type for genes associated with innateness gradient. The upper panel shows top 100 positive and negative significant associations within protein-coding genes. The lower panel shows significant associations for 92 lncRNA genes. Genes within each heatplot were sorted by β. Gene expression level was scaled by row. N = 6 donors, 1–2 replicates per cell type. Boxplots are described in Methods

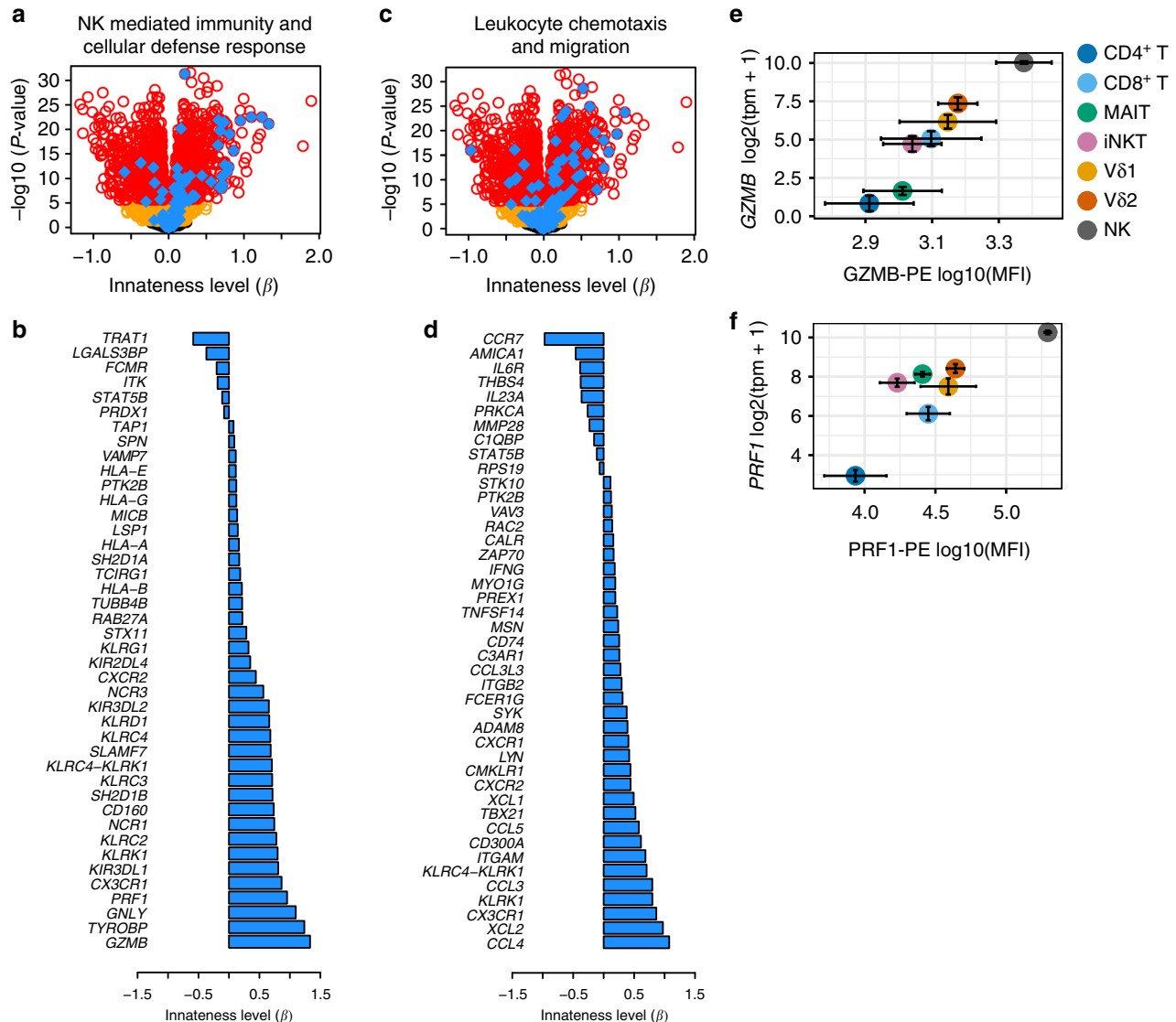

**Fig. 3** Genes and pathways associated with innateness. **a** Volcano plot showing associations with the innateness gradient. Yellow, genes with $P < 0.05$; red, genes with $P < 2.5\mathrm{e}{-06}$ (Bonferroni threshold); blue, genes with GO terms involving NK-mediated immunity and cellular defense response (GO:0006968, GO:0002228). **b** Innateness level ($\beta$) for individual genes in **a** with $P < 2.5\mathrm{e}{-06}$. **c** As in **a** but with blue showing genes from GO terms involving leukocyte chemotaxis and migration (GO:2000501, GO:0035747, GO:1901623, GO:0030595, GO:2000401, GO:0097530, GO:0097529, GO:0048247, GO:0072676, GO:1990266). **d** Innateness level ($\beta$) for individual genes in **c** with $P < 2.5\mathrm{e}{-06}$. **e** *GZMB* and **f** *PRF1* flow-cytometric validation, showing protein levels (x-axis), and transcript levels with RNA-seq (y-axis)

Methods section). All subsequent *P*-values reported for association with the innateness gradient are derived from linear mixed models, likelihood ratio test), including protein coding and lncRNA genes (Fig. 2c, Supplementary Data 3). Hereafter, we refer to positive and negative associations with the ranked gradient as associations with "innateness" and "adaptiveness," respectively. We quantified the level of innateness as the magnitude of the change in expression level by an increase of one in the gradient (the $\beta$ of the gradient variable within our linear mixed model).

**Associations with innateness**. The Gene Ontology (GO) terms most associated with innateness included NK cell and lymphocyte chemotaxis, NK cell-mediated immunity, cellular defense response, and several additional terms related to leukocyte migration and activation (Fig. 3a–d, specific GO terms indicated in figure legend and Supplementary Figure 5a). Using flow

cytometry, we validated the expression of key genes, including killer cell lectin-like receptor (KLR) family genes and killer cell immunoglobulin-like receptor (KIR) genes (Supplementary Figure 5b). Cytotoxicity proteins, such as perforin, granzyme B, and granulysin also associated with innateness (Fig. 3e, f, Supplementary Figure 5b). Eight chemokines strongly associated with innateness, including *CCL3*, *CCL4*, *CCL5*, *XCL1*, and *XCL2* ($P < 9\mathrm{e}{-12}$), consistent with a role for innate lymphocytes in recruiting other inflammatory cell types to initiate inflammation.

*IFNG* (the gene coding for IFN-γ) showed a significant association with innateness ($P = 1.7\mathrm{e}{-06}$, Fig. 4a), and the baseline *IFNG* levels in each cell population predicted their production of IFN-γ upon stimulation (Supplementary Figure 2a, b). Since ITC produce diverse cytokines and chemokines[1,3], we quantified the total cytokine and chemokine transcriptome "mass" in each cell type at baseline. We observed that the aggregate sum of the expression levels of the 37 cytokines and chemokine genes expressed in our dataset followed the innateness

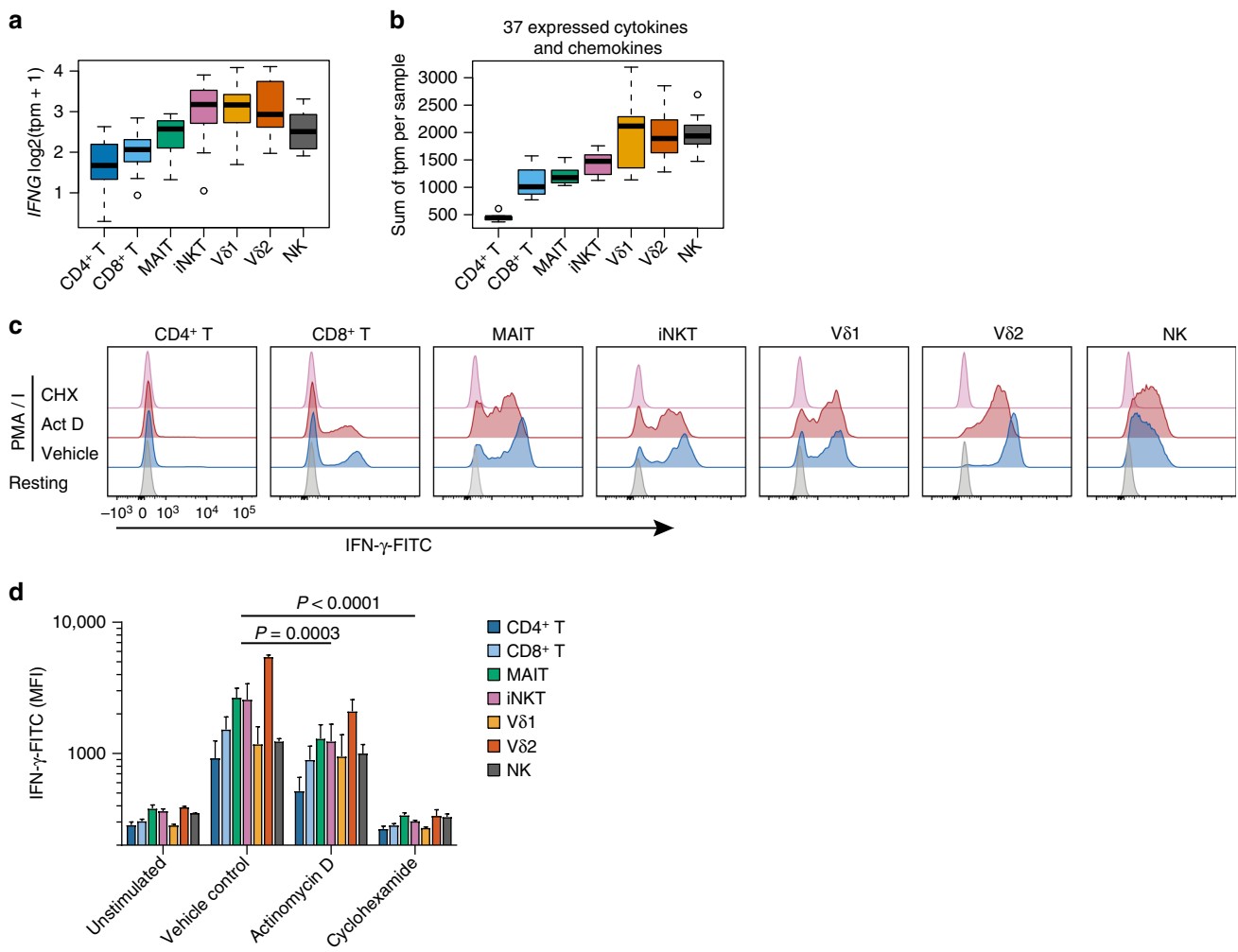

**Fig. 4** Cytokine and chemokine mRNA in ITC. **a** Distribution of *IFNG* transcript levels across all samples. **b** Sum of expression levels (tpm) for 37 cytokine and chemokine genes across all samples, per cell type. tpm, N = 6; Boxplots are described in Methods. Histogram overlays are normalized to mode. **c** Representative flow cytometry plots for IFN-γ intracellular staining 2 h after activation in the presence of cyclohexamide to block transcription or actinomycin D to block translation. **d** As in **c**, with MFI shown for N = 3, error is s.e.m., *P*-value is paired *t* test

gradient (Fig. 4b). Although we could detect mRNA for cytokines, intracellular staining did not reveal baseline cytokine protein production in ITC or adaptive T-cell populations. We hypothesized that preformed mRNAs might contribute to rapid cytokine production across ITC. To test this hypothesis, we activated PBMCs with PMA and ionomycin for 2 h in the presence of actinomycin D, which blocks new transcription, or cycloheximide, which blocks translation. Early IFN-γ production by all lymphocyte subsets was nearly completely blocked by cyclohexamide, but only partially blocked by actinomycin D (Fig. 4c, d). These experiments show that preformed mRNA contributes to early cytokine production by ITC, adaptive T cells, and NK cells. Our data suggest that translation of preformed effector mRNA may be one of the mechanisms that enable the characteristic rapid response of ITC populations.

Metabolic pathways are well-known to vary among immune cell subsets and influence their functions[13]. Among metabolic programs, the pentose phosphate pathway was nominally positively associated with innateness (*P* = 0.036, gsea() liger package, Supplementary Figure 6a). *G6PD*, the gene that codes for the rate-limiting enzyme in the pentose phosphate pathway, showed the strongest positive association with innateness in this pathway (β = 0.29, *P* = 3e–14, Supplementary Figure 6b, c). This

enzyme produces NADPH, which in turn can be used for glutathione biosynthesis, protecting against damage caused by ROS. Two critical enzymes for buffering the damaging effect of ROS, *GCLM*, and *GCLC*, also nominally associated with innateness (*P* = 2e–04 and 1e–03, respectively, Supplementary Figure 6d,e). We quantified ROS by flow cytometry using CellROX green, and found that total cellular ROS levels were higher in adaptive T cells than in ITC, suggesting that elevated *G6PD* might provide a baseline buffer counteracting ROS (Supplementary Figure 6f, g). Overall, these results suggest that ITC are prepared to buffer ROS at baseline, a useful adaptation for effector cells expressing chemokine receptors, such as CCR1, CCR2, and CCR5 (Supplementary Figure 9) that direct them to the same sites of infection or inflammation as monocytes and neutrophils.

**Associations with adaptiveness.** When we applied gene set enrichment to adaptiveness, "cytosolic ribosome" (GO:0022626) emerged as the most-associated term (*P* = 4.7e–28, hypergeometric test, Fig. 5a, b, Supplementary Figure 7a, b). This enrichment was not driven by a small percentage of genes very strongly overexpressed among ITC (Supplementary Figure 7c).

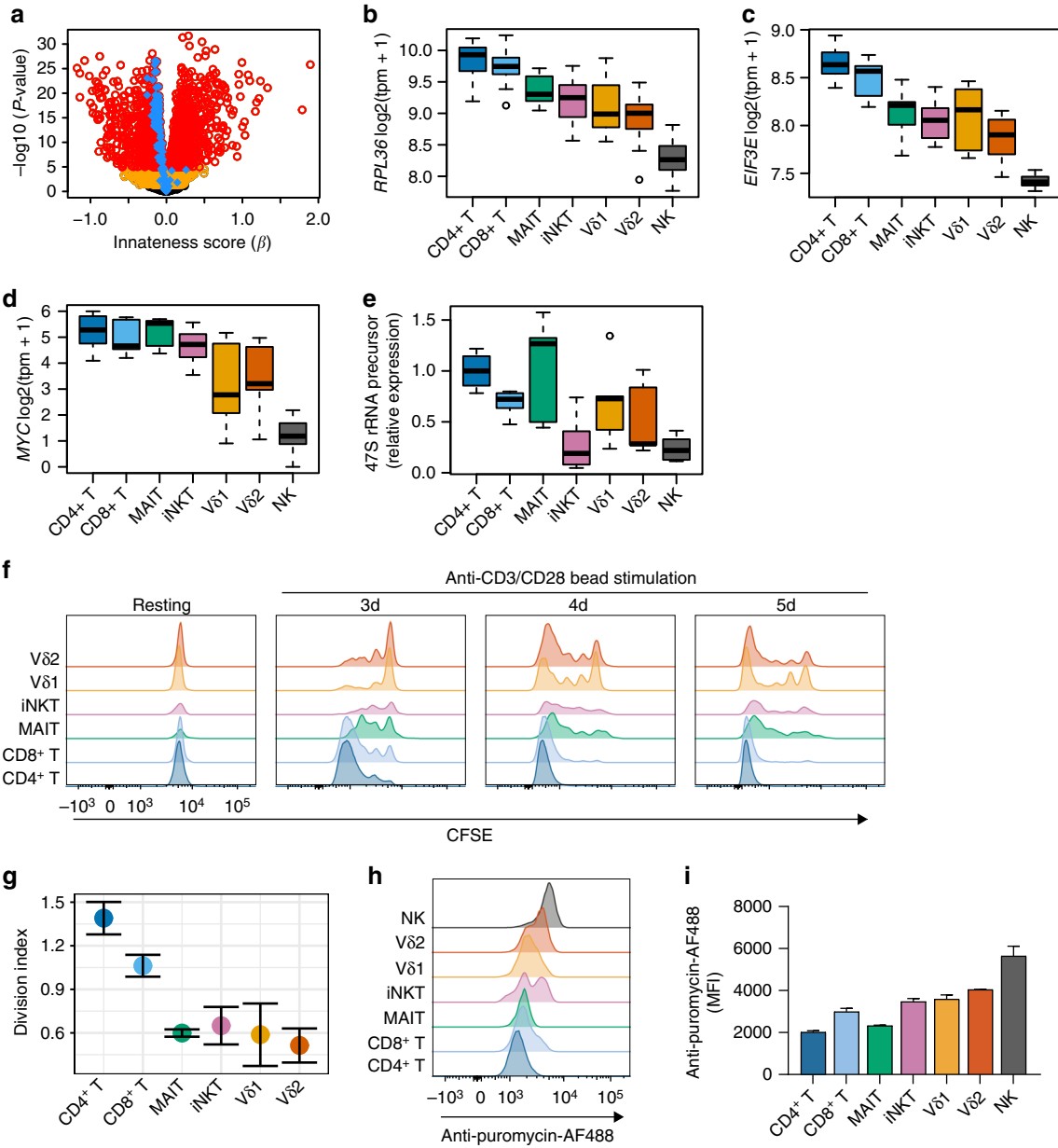

**Fig. 5** Reduced ribosome transcripts and proliferation in ITC. **a** Volcano plot showing results of associations with the innateness gradient. Yellow, genes with P < 0.05; red, genes with P < 2.5e−06 (Bonferroni threshold); blue, GO term cytosolic ribosome (GO:0022626). Distribution by cell type for transcript levels of **b** ribosomal protein *RPL36*, **c** translation initiation factor *EIF3E*, and **d** *MYC* (N=6). **e** qPCR for 47 S rRNA precursor from sorted cell populations (N = 3). Expression shown is relative to *HPRT*, and normalized to CD4+ T cells for comparison between samples. **f** PBMCs were labeled with CFSE and then cultured with anti-CD3/CD28-coated beads before staining with markers to identify ITC populations, **g** division index shown at day 3 (N = 3, s.e.m.). Ribopuromycylation in PBMCs to quantify total translational activity measured by flow cytometry, **h** representative plot, and **i** N = 3, s.e.m. Boxplots are described in Methods. Histogram overlays (**g**, **h**) are normalized to mode

Translation initiation factors were also consistently associated with adaptiveness (Fig. 5c, Supplementary Figure 7d), suggesting that the translational machinery, and not just the ribosome complex, was associated with adaptiveness. *MYC*, which coordinately regulates ribosomal RNA genes[14], was the transcription factor with the highest fold change associated with adaptiveness (P = 3.8e−22, Fig. 5d). As an independent assessment of ribosome synthesis, we used quantitative polymerase chain reaction (qPCR) to assess expression of the earliest uncleaved ribosomal RNA (rRNA) precursor. The expression of precursor 47S rRNA associated with adaptiveness (Spearman rho = −0.57, P = 9e−05,

t test, Fig. 5e), suggesting that ITC have a relative decrease in ribosome biogenesis.

Since new ribosome production is necessary for proliferation, and *MYC* expression is generally associated with proliferative capacity, we hypothesized that proliferation potential might associate with adaptiveness. We assayed proliferation in primary human T cells in response to anti-CD3/CD28-coated beads, by measuring carboxyfluorescein succinimidyl ester (CFSE) dye dilution. NK cells were omitted from this analysis, since they do not respond to anti-CD3/CD28-coated beads. Like *MYC* and ribosome biogenesis, proliferation was associated with

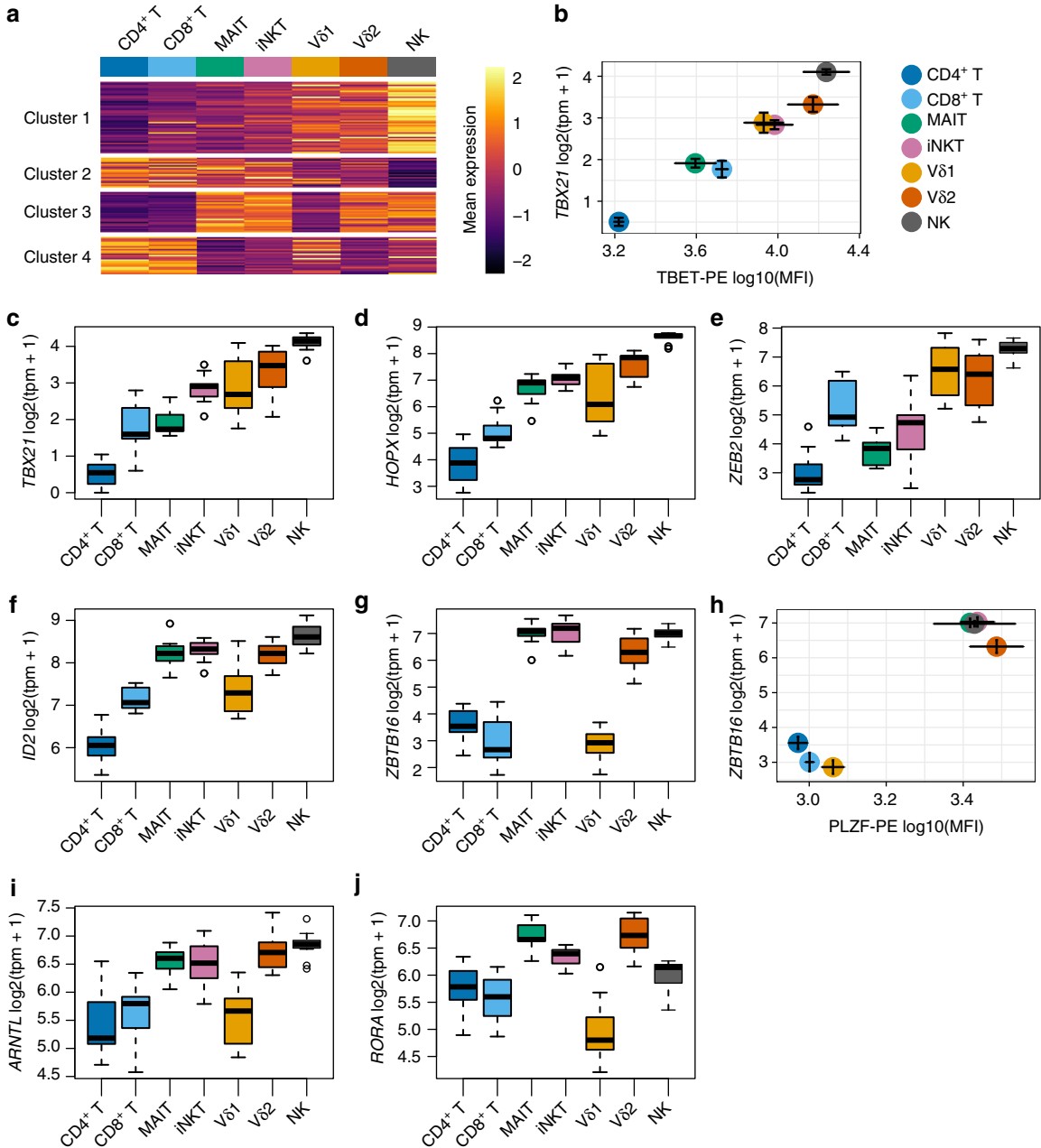

**Fig. 6** Transcription factors in ITC. **a** Heatmap showing mean expression levels (scaled by row) for variable transcription factors among cell types, clustered into four groups. **b** Flow-cytometric quantification of T-bet ($N = 3$) compared to transcript levels with RNA-seq for its encoding gene *TBX21* ($N = 6$). Cluster 1, **c** *TBX21*, **d** *HOPX*, and **e** *ZEB2*; Cluster 3, **f** *ID2*, **g** *ZBTB16*, **i** *ARNTL*, and **j** *RORA*. **h** Flow-cytometric quantification of PLZF ($N = 3$) compared to transcript levels for its encoding gene *ZBTB16* ($N = 6$). Error for tpm vs. MFI is s.e.m. Boxplots are described in Methods

adaptiveness and innate T cells proliferated less than adaptive T cells (Spearman rho $= -0.73$, $P = 5.8e{-}04$, $t$ test, Fig. 5f, g). These results recall the well-described regulation of ribosomes in prokaryotes, where ribosome biogenesis is the major energetic control point, is suppressed in conditions under which growth and division are deprioritized[15], and can be fine-tuned to ensure maximal occupancy of active ribosomes[16].

To address the possibility that innate cells might be translationally quiescent, we assayed ribopuromycylation to quantify total active translation[17]. Ribopuromycylation was positively associated with innateness, suggesting that innate cell types are engaged in more active translation than the adaptive T cells at baseline (Fig. 5h, i). This suggested that despite having

lower expression of the ribosomal mRNA and rRNA, innate T cells were not translationally quiescent, but rather that they deprioritized ribosome generation. Consistent with this model, RNA polymerase I component *POLR1D*, which is responsible for rRNA transcription, was associated with adaptiveness. On the other hand, RNA polymerase II components *POLR2G*, and *POLR2K*, which transcribe mRNA, were significantly associated with innateness (all $P < 8e{-}07$, Supplementary Figure 8). Taken together, these results suggest that adaptive cells use their ribosomes to make more ribosomes, thus prioritizing the production of factors required for cell growth and division, while innate cells may suppress transcription of ribosomal genes to optimize the usage of available ribosomes for other RNAs, such as

mRNAs encoding effector functions, including the rapid production of cytokines (Fig. 4).

**Transcriptional regulation of innateness**. We identified 142 transcription factors that varied significantly between cell types (P < 5.8e−05, $f$ test aov() R, Bonferroni threshold). The expression of these transcription factors across cell types clustered into four major groups (Fig. 6a). Cluster 1 showed a gradual increase that closely matched the pattern of the innateness gradient. Cluster 2 showed a pattern opposite to that of cluster 1, with an increase in expression toward adaptive cellular populations. Cluster 3 showed high levels of expression in iNKT, MAIT, Vδ2, and NK cells, with relatively lower levels in adaptive T and Vδ1 T cells, and cluster 4 captured transcription factors with the opposite pattern to cluster 3 (Fig. 6a). In PCA of these transcription factors, the second principal component separated iNKT cells, MAIT, and Vδ2 T cells, from adaptive and Vδ1 T cells (Supplementary Figure 9a), similar to K-means clusters 3 and 4. These same cell groupings were also captured by PC2 generated using the overall most variable genes (Fig. 2a).

Within cluster 1 of innateness-associated transcription factors, T-bet (*TBX21*, P = 2.4e−29), known for important roles in type 1 helper T cell (Th1) and iNKT-cell effector functions[18,19], followed the innateness gradient at both the transcript and protein levels (Fig. 6b, c). The next two innateness-associated transcription factors with the highest fold changes were *HOPX* and *ZEB2* (Fig. 6d, e). *HOPX* (P = 7.2e−25), reported to be induced by T-bet, has been shown to regulate persistence of effector memory Th1 cells, with upregulation in terminally differentiated cells[20]. *ZEB2* (P = 1.8e−18) has been reported to cooperate with T-bet to induce terminal differentiation of cytotoxic T lymphocytes[21,22]. Two NFAT family proteins, *NFATC2* (P = 2e−16) and *NFAT5* (P = 1.1e−9), were associated with cluster 1 transcription factors. *IRF8* (P = 3.1e−14), *TFDP2* (P = 5.9e−13), *NFIL3* (P = 2.7e−13), *KLF10* (P = 2.1e−15), *RUNX3* (P = 5.6e−18), *LITAF* (P = 6.2e−24), *ZSCAN9* (P = 9.4e−17), and *ZNF600* (P = 4e−17) were also in this cluster and significantly associated with innateness. Within the adaptiveness-associated cluster 2, besides *MYC* (Fig. 5d), we found *TCF7*, involved in the maintenance of T-cell identity (P = 2.4e−21)[23], *BACH2* (P = 4.3e−10), *NR3C2* (P = 1.8e−10), *POU6F1* (P = 1.9e−10), and *BCL11B* (P = 2e−17).

The third cluster of innateness-associated transcription factors, those enriched in iNKT cells, MAIT, Vδ2 T, and NK cells, included *ID2* (P = 8.8e−13, Fig. 6f), *MYBL1* (P = 1.52e−10), *BHLHE40* (P = 8.1e−11), *FOSL2* (P = 8.8e−14), and *ZBTB16* (P = 1.1e−5, Fig. 6g, h). Among these genes, *BHLHE40*, *FOSL2*, *ZBTB16* (encoding PLZF), and *ID2* have been reported to contribute to iNKT-cell development and/or activation in mice[24–28]. Id2 is also a major regulator of ILC development[29], and has been implicated in the regulation of mouse iNKT[30], ILC1[31], and CD8[+] T cell[32] effector functions in the periphery. Published transcriptional profiles of NK cells, ILC1, and influenza-specific Id2-deficient mouse CD8[+] T cells showed a striking concordance of Id2-dependent expression with our innateness gradient genes, highlighted by *TBX21*, *ZEB2*, *IL18RAP*, *CCR7*, *TCF7*, cytotoxicity, and KLR genes[31–33]. *TCF7*, consistently downregulated in ITC, is negatively regulated by Id2, suggesting that in part, Id2 may drive the loss of adaptive T-cell identity observed in ITC. Taken together, these data suggest that Id2 may drive many features of innateness in human ITC, and may be a major transcriptional node involved in maintaining their baseline innate state.

PLZF is a zinc finger transcription factor known to be important for the development and function of iNKT cells[34,35], MAIT cells[35], and innate lymphoid cells[36]. Mean PLZF protein expression by intranuclear staining confirmed our mRNA

expression results (Fig. 6h). Human γδ T cells have previously been reported to express PLZF[37], but we did not detect elevated PLZF expression in Vδ1 cells (Fig. 6g, h). Differential expression analysis between PLZF[+] ITC and adaptive T cells revealed "cytokine receptor activity" as the most enriched term for upregulation in PLZF[+] ITC (P = 7.9e−05, hypergeometric test). PLZF expression in T cells was also associated with the aggregate expression of all cytokine and chemokine receptor activity genes (Supplementary Figure 9b), and we validated the expression of several of these receptors by flow cytometry (Supplementary Figure 9c). We tested migration across a permeable membrane to a panel of chemokines, including CCL19 that directs lymphoid recirculation by CCR7 and CCL2, 3, 4, and 8 targeting chemokine receptors shared between ITC (Supplementary Figure 9c) and myeloid cell populations, such as monocytes. PLZF[+] ITC readily migrated in response to classical monocyte chemokines, and also in response to CCL19, while migration of adaptive T cells and the PLZF[−] Vδ1 population was limited to CCL19 (Supplementary Figure 10).

For genes differentially expressed between PLZF[+] ITC and adaptive T cells, we found significant enrichment of PLZF target genes identified in mouse thymocytes with ChIP-seq[38] (P = 6.2e−07, $χ^2$ test, Supplementary Figure 9d). In addition, PLZF[+] ITC upregulated genes that were associated with the term "circadian regulation of gene expression" (P = 4.2e−04, hypergeometric test), with major clock transcription factor genes like *ARNTL* (that codes for BMAL1), *RORA*, *PER1*, and *CRY1* significantly upregulated in PLZF[+] ITC compared with adaptive T cells (P < 5e−08, linear mixed models, likelihood ratio test) (Fig. 6i, j, Supplementary Figure 9e). Both *BHLHE40* and *ID2* also have the capacity to regulate the circadian clock[39–41]. Notably, although human NK cells express PLZF (mature mouse NK cells do not express PLZF), many genes upregulated in PLZF[+] ITC and identified as PLZF targets in mouse[38] showed low expression in human NK cells, including *CCR2*, *CCR7*, *CXCR6*, *RORC*, *CCR5*, *CCR6*, and *LTK* (Supplementary Figure 9c, d). These results suggest that PLZF may regulate different sets of genes depending on the cell type, likely working as part of a larger gene network in determining ITC fate.

**Innateness in other ITC populations**. We next investigated the innateness gradient in other candidate innate-like human T-cell subsets. We chose two additional T-cell populations for analysis, Vδ3-expressing γδ T cells and δ/αβ T cells, each of which can constitute up to 1% of human peripheral T cells[42,43]. We sorted Vδ3 T cells and δ/αβT cells in duplicate from one individual, and profiled their transcriptomes with ultra-low- input RNA-seq. δ/αβ and Vδ3 clones have been identified that, like iNKT cells, recognize α-galactosylceramide presented by CD1d[42,43], suggesting that these cells might potentially play a similar role in immunity to iNKT cells. However, PCA revealed that δ/αβ T cells were closer to adaptive T cells, and closest to CD8[+] T cells, rather than segregating with iNKT cells and other innate T cells (Supplementary Figure 11). This suggests that δ/αβ T cells may have an adaptive-like phenotype. Vδ3 T cells, on the other hand, segregated closer to innate T cells by PCA, among the other γδ T cells (Supplementary Figure 11). Neither δ/αβ T cells or Vδ3 T cells expressed PLZF.

**Innateness in adaptive populations**. Cytotoxicity genes and NK markers are expressed by a subset of adaptive T cells. We found that this class of genes was expressed by CD8[+] T cells, and in some cases at higher levels than in ITC. Interestingly, the development of innate-like Th1 effectors from adaptive cells has also recently been demonstrated in mice[44]. To assess expression of

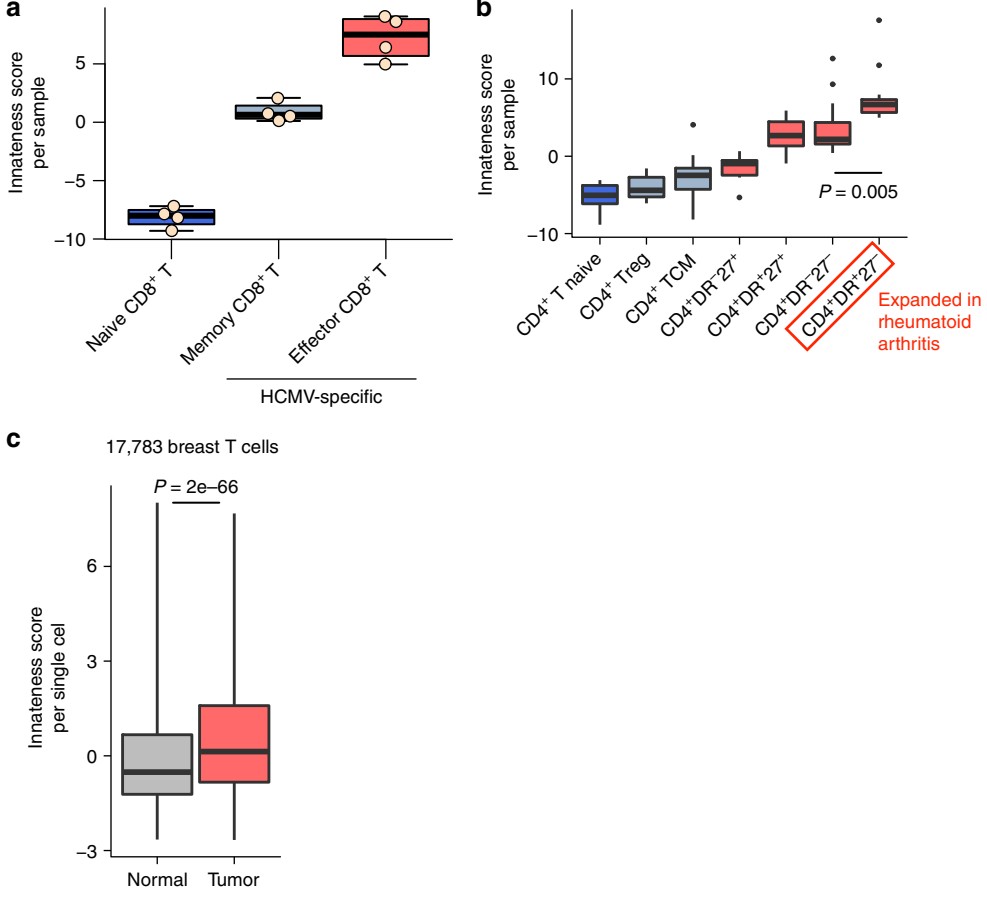

**Fig. 7** Innateness in adaptive T-cell populations. Analysis of innateness score (based on PC1 loadings) for transcriptomic datasets from **a** HCMV-specific CD8+ T-cell populations (naive = CD45RA+CD27bright; memory = CD45RA−CD27+; effector = CD45RA+CD27−), and **b** CD4+ T-cell populations from patients with rheumatoid arthritis or osteoarthritis (TCM = T central memory; red boxplots indicate effector subsets). Boxplots are described in Methods. **c** Single-cell RNA-seq imputed expression data from T cells from normal or tumor breast tissue with innateness score calculated for each cell. Boxes show the first to third quartile with median, whiskers encompass 95% of the data. **a** N = 4 replicates, **b** N = 7 rheumatoid arthritis and 6 osteoarthritis donors, **c** N = 17,783 breast T cells from eight cancer patients. P-values are Wilcoxon test

innateness gradient genes in human adaptive effector T cells, we re-analyzed a human expression dataset generated using MHC class I tetramer-sorted, HCMV-specific CD8+ T cells[45] (polyclonal human CD8+ T-cell datasets would likely be substantially "contaminated" with ITC). To quantify the total level of innateness in each sample, we generated an "innateness score." For this metric, we used the PC1 weights (loadings) for genes included in our PCA (Fig. 2a) and multiplied them by the expression levels of these genes in the query dataset. This integrates the signals that come from both innateness and adaptiveness genes into a single score, which essentially reflects a projection into our PC1 (Fig. 2b). HCMV-specific effector memory CD8+ T cells had a higher innateness score than HCMV-specific memory CD8+ T cells, which in turn had a higher score than naive CD8+ T cells (Fig. 7a). The same patterns are captured if we look at expression levels of innateness genes, and the opposite trend is observed with expression levels of adaptiveness genes (Supplementary Figure 12a, b).

We also analyzed published RNA-seq data for CD4+ T-cell subsets from healthy individuals[46]. CD4+ effector memory T cells had a higher innateness score than CD4+ central memory T cells (P = 0.008, Wilcoxon test), which had a higher innateness score than CD4+ naive T cells (P = 0.032, Wilcoxon test, Supplementary Figure 12c–e). To investigate innateness in CD4+ T cells during inflammation, we calculated the innateness score for

CD4+ T-cell populations profiled by low-input RNA-seq from patients with arthritis[47], which is thought to be driven in part by CD4+ T cells[48]. The innateness score ordered T-cell populations from naive to central memory to effector cells (Fig. 7b). The subset that scored the highest in innateness, CD4+HLA-DR+CD27− T cells, are the precise subset that is most expanded in rheumatoid arthritis and correlate with treatment response[47].

We next interrogated lymphocytes from a single-cell RNA-seq dataset from patients with breast carcinoma[49]. As seen for HCMV-specific CD8+ T cells (Fig. 7a) and CD4+ T cells from rheumatoid arthritis (Fig. 7b), lymphocyte populations from patients with breast carcinoma, as annotated in the original study, recapitulated the innateness gradient (Supplementary Figure 13). Further, T cells from within the tumor showed a higher innateness score than those in healthy tissue (P = 2e−66, Wilcoxon test), suggesting that T-cell innateness was enriched at the site of disease (Fig. 7c).

**Heterogeneity in ITC and adaptive populations.** To assess heterogeneity among the lymphocyte populations studied, we applied single-cell RNA-seq to the same seven cell populations studied above. We first sorted CD4+ T, CD8+ T, iNKT, MAIT, Vδ1, Vδ2, and NK cells from two donors and then labeled each population with DNA-barcoded antibodies, allowing us to

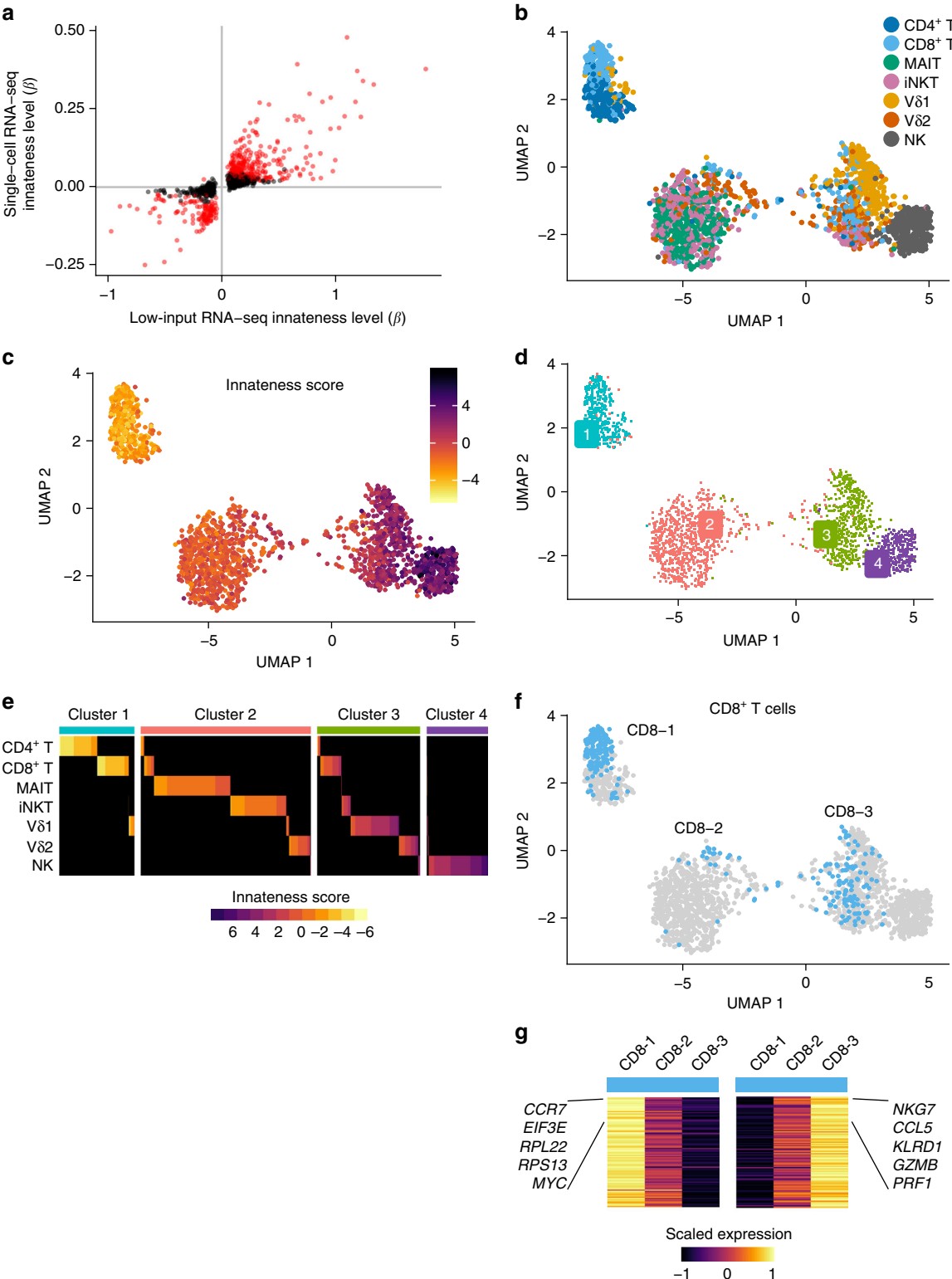

**Fig. 8** Single-cell RNA-seq of ITC. In total, 2306 lymphocytes sorted and barcoded with antibodies to confirm population identity. **a** Level of innateness ($\beta$) in low-input RNA-seq vs. single-cell RNA-seq for genes that were significantly associated with innateness gradient in low-input RNA-seq. In red, genes that were also significant in single-cell RNA-seq (Bonferroni threshold). **b** UMAP colored by cell type as identified with hashing antibodies, or **c** innateness score calculated for each cell, or **d** by the four clusters identified. **e** Heatplot showing cells colored by innateness score, separated by cell type as identified by hashing antibodies (rows), and sorted by the cluster and innateness score. **f** UMAP showing CD8+ T-cell subpopulations in clusters 1, 2, and 3. **g** Left panel, genes upregulated in CD8-1 vs. CD8-2 and CD8-3; right panel, genes upregulated in CD8-3 vs. CD8-1 and CD8-2. Heatmap shows mean expression levels (scaled by row). $N = 2$ donors per sorted cell population

identify the original cell populations using a cell "hashing" protocol[50]. We then assayed all populations together in a single-pooled droplet-based assay, which obviated any batch effects that might have occurred if the populations were assayed separately[51]. After stringent quality control, we obtained 2036 high-quality single-cell libraries with similar representation of each starting cell population (Supplementary Table 2, Supplementary Figure 14).

Before exploring heterogeneity, we wanted to ensure that the broad trends seen in the bulk RNA-seq data were replicated in the single-cell RNA-seq data. We first compared the single-cell and low-input RNA-seq datasets on a gene-by-gene basis. From our single-cell data, we calculated the level of innateness ($\beta$ based on the association with innateness rank order determined in low-input RNA-seq) for the significant adaptiveness and innateness genes identified in the low-input RNA-seq data. Unsurprisingly, the single-cell gene expression associations with the innateness gradient replicated our low-input RNA-seq data (Fig. 8a, Spearman rho = 0.82, 94% concordance in direction of effects). Key innateness genes, such as *GZMB*, *PRF1*, *KLRD1*, *NKG7*, *HOPX*, *ZEB2*, and *ID2*, and adaptiveness genes, such as *RPL36*, *RPL22*, *RPS19*, *EEF3E,* and *MYC* were also significantly associated with innateness in single-cell expression data (all $P < 7e{-}18$, linear model, likelihood ratio test).

For dimensionality reduction and data visualization, we applied PCA to the single-cell data using the top 1545 variable genes. PCA of single-cell RNA-seq data (sc-PC1 and sc-PC2) mirrored those obtained for low-input RNA-seq, with sc-PC1 separating adaptive cells from innate cells. Among the innate populations, NK cells were on one end, iNKT and MAIT cells on the other end, and γδ T cells in-between (Supplementary Figure 15a). PC1 from the single-cell data (sc-PC1, Supplementary Figure 15b) largely recapitulated the innateness gradient derived from low-input RNA-seq (Fig. 2b), with the position of Vδ1 and Vδ2 reversed. The innateness order for Vδ1 and Vδ2 was different between the two donors included in the single-cell study, suggesting possible donor-to-donor variability among γδ T cells (Supplementary Figure 15c). The innateness score, calculated for each single cell using PC1 loadings from the low-input bulk RNA-seq data was highly correlated with sc-PC1 (Pearson correlation $r = 0.9$, Supplementary Figure 15d).

To better understand cellular heterogeneity, we embedded and clustered single-cell data. We embedded single-cell data using the top 20 principal components into a two-dimensional space with uniform manifold approximation and projection (UMAP). Adaptive T cells, followed by ITC and NK cells were spread across the UMAP (Fig. 8b). Key innateness and adaptiveness genes colored a gradient across the UMAP (Supplementary Figure 16), as did the innateness score calculated per cell (Fig. 8c). To define underlying innateness states at the single-cell level, we applied shared nearest-neighbor modularity clustering which identified four distinct clusters. We propose that these groups represent four "states of innateness" that can be adopted by lymphocytes. The most adaptive of the four clusters, Cluster 1 contained mostly CD4+ T and CD8+ T cells. Cluster 2 contained mainly iNKT, MAIT, and Vδ2 T cells. Cluster 3, was even more innate, and contained predominantly Vδ1 and Vδ2 T cells. Finally, Cluster 4 was completely innate, consisting mostly of NK cells (Fig. 8d).

For some cell types, we noted that sizeable subsets were present in multiple clusters (Fig. 8e, Supplementary Figure 17). CD8+ T cells exhibited the most striking heterogeneity, showing populations that clustered into three different states: CD8-1 clustered with adaptive CD4+ T cells, CD8-2 that was most similar to iNKT and MAIT cells, and CD8-3 cells that were most similar to γδ T cells (Fig. 8f). When we examined the genes that

separated each CD8+ T-cell cluster, we identified genes that were part of the innateness and adaptiveness gene sets identified in our low-input RNA-seq data (182 out of 237 unique genes that were significant in any of the three one-vs.-all differential expression comparisons, $P < 4.4e{-}05$, linear model, likelihood ratio test, Bonferroni threshold). The CD8-2 population was characterized by intermediate expression of innateness and adaptiveness genes, rather than by expression of unique genes (Fig. 8g). Only four genes, *GZMK*, *TMSB10*, *CCL5*, and *GMC1* were significantly differentially expressed in the CD8-2 population. Vδ1 T cells were predominantly in Cluster 3 with Vδ2 T cells, although a substantial Vδ1 population also populated Cluster 1 with adaptive CD4+ T cells (Supplementary Figure 17). Ninety-four percent of genes that distinguished this adaptive-like Vδ1 population ($P < 5.2e{-}05$, linear model, likelihood ratio test, Bonferroni threshold) were adaptiveness and innateness genes, including downregulated cytotoxic genes, such as *PRF1* and *KLRD1*, and upregulated ribosomal machinery genes, such as *RPL34*, *RPL22*, and *EEF1A1*. This finding is consistent with recent reports on Vδ1 populations being highly variable across donors, and with dynamic changes in response to infection, suggesting that a naive Vδ1 state may be part of their development[52,53]. Besides NK cells, and MAIT cells, most of the lymphocytes assayed spanned multiple clusters. Taken together, our single-cell RNA-seq data demonstrate that the innateness cell-type hierarchy seen in low-input RNA-seq reflects an average of multiple cell states, and that individual innate and adaptive T populations variably populate these innateness states.

## Discussion

MAIT, iNKT, γδ, and other innate-like T cells do not fit neatly into traditional paradigms of adaptive or innate immunity. Each population has been studied in depth individually, but rarely have they been considered in aggregate. Here, we set out to study human ITC as a group, addressing two important questions: (1) is there a shared transcriptional basis for their functions in immunity, and (2) how do ITC maintain their baseline "poised" effector state? In quantitative, unbiased analyses using both low-input RNA-seq and single-cell RNA-seq, we discovered that ITC segregate along an innateness gradient between prototypical adaptive and innate populations. We propose that the large transcriptional programs positively and negatively associated with this gradient represent the transcriptional basis of lymphocyte innateness. Though they share the same lineage as adaptive T cells, our data support that ITC are indeed a "family" in a sense, with a common transcriptional basis for their similar functions in immunity, including rapid cytokine and chemokine production, chemotaxis to areas of inflammation, cytotoxicity, and TCR-independent responses. The functional and transcriptional conservation of innate-like functions in ITC suggests that they enhance evolutionary fitness. That humans dedicate such a large part of their T-cell repertoire to the generation of innate-like receptors is a testament to the teleological importance of innate immune surveillance even after the evolution of adaptive immunity.

Strikingly, we observed that this innateness program could also differentiate adaptive effector populations in health and disease. Analysis of published datasets demonstrated that naive, memory, and effector-adaptive populations could be classified by their innateness. Thus, the study of ITC highlights important pathways used across innate and adaptive lymphocyte populations. Consistent with this notion, our single-cell RNA-seq data identified populations of CD8+ T cells and even some CD4+ T cells that clustered within the two "innate states" that largely represent ITC populations. The genes that differentiated these populations were

the same genes identified in the innateness gradient. These data suggest that lymphocytes can exist in defined states of innateness whether achieved developmentally, as is thought to be the case with ITC, or through experience, in the case of adaptive T cells.

The shared transcriptional programs associated with innateness included cytokine/chemokine production, cytotoxicity, and cytokine/chemokine receptor expression. For the genes positively associated with the innateness gradient, this is essentially an "effector gradient," which strongly supports a role for ITC in host defense. We found that human ITC rapidly produced IFN-γ after activation through their TCRs (Supplementary Figure 2), as do a smaller fraction of adaptive T cells. However, IFN-γ production in response to IL-12, IL-18, and IFN-α, cytokines generated by myeloid or stromal cells in response to danger signals, were almost exclusively limited to ITC (Fig. 1b). This is consistent with the role of ITC as innate responders where prior pathogen experience is not required. Thus, T-cell innateness can regulate the response to pathogen-associated molecular patterns. Of note, human Vδ1 cells have been demonstrated to be variable in both TCR repertoire and numbers, and likely respond to specific infections[52,53]. Although they express much of the innateness program, Vδ1 cells may not fit the ITC paradigm as neatly as the more-conserved MAIT, iNKT, and Vδ2 populations. Indeed, Vδ1 cells have greater TCR diversity, exhibit less cytokine-only activation, do not express PLZF, and a portion of these cells clustered with adaptive T cells in our single-cell RNA-seq analysis.

Our identification that mRNAs encoding for ribosome subunits and other factors involved in translation associated with adaptiveness (Fig. 5) also sheds light on what it means to be innate. In a given cell, loss of rapid proliferative capacity may be an inherent trade-off for enhanced effector function, a balance previously observed for CD8+ T cells[54]. For an adaptive T cell, population expansion is of central importance during immune responses. ITC, on the other hand, are likely to function as sentinels early during infection, acting to enhance ensuing immune responses in response to microbial molecules. For such a role, rapid effector responses are key, and proliferation may serve only to replenish numbers at a later stage. Taken together, the effector-focused transcriptional programs of ITC and proliferation-focused programs of adaptive cells are ideally suited to support their respective roles in immunity. It is notable that effector gene programs in ITC and proliferative gene programs in adaptive cells were present at rest. Our data showing that a substantial portion of early IFN-γ is produced from preformed mRNA (Fig. 4c, d), together with higher overall active translation in ITC (Fig. 5i), support the concept of ITC being in a poised state ready for a rapid and efficient effector response. Similarly, we postulate that adaptive cells are poised for division, with relatively higher RNA polymerase I, MYC expression, and prioritization of ribosome formation that would be needed for cell proliferation.

Finally, the innateness gradient reported here could be applied in different scenarios in order to better understand human immunology and human disease. A transcriptomic innateness score could be employed as a unified T-cell metric to classify individual single cells assayed with single-cell RNA-seq, providing a better understanding of patient heterogeneity. We provide data that innateness in the T-cell compartment increases with infection, inflammation, and in cancer. The "innateness score" (Fig. 7) may be a useful prognostic indicator, and understanding the molecular mechanisms associated with innateness may pave the way for novel therapies aimed at modulating this coordinated transcriptional response. We can also use our immunoprofiling data and create an "individual innateness metric" for each individual based on the abundance of each T-cell type weighted by the innateness level of that cell type. This score is remarkably variable between individuals, even after correcting for age (Supplementary Figure 18). This single innateness metric in an individual might be associated with genetic differences, human diseases, including cancer, infection, and allergy, or therapeutic responses to immunomodulating medications.

## Methods

**Glossary of innateness terms.** "Innateness" and "adaptiveness" genes are those that in the low-input RNA-seq were significantly associated with the rank order of each lymphocyte population in the innateness gradient (CD4+ T = 1, CD8+ T = 2, MAIT = 3, iNKT = 4, Vδ1 = 5, Vδ2 = 6, and NK = 7).

"Innateness level" is the magnitude of the change in expression level by an increase of one in the gradient variable (the β of the gradient variable within our linear mixed model in low-input RNA-seq or linear model in single-cell RNA-seq).

"Innateness score" is calculated per sample or per single-cell, by selecting the genes used in the low-input PCA (Fig. 2a), multiplying their PC1 loading by the scaled-expression value on the query sample, and summing all up. This integrates the signals that come from both innateness and adaptiveness genes into a single score, which essentially reflects a projection into our PC1.

"Individual innateness metric" is calculated per individual by integrating the immune profiling data with the innateness gradient rank per cell type. Specifically, we summed the abundance per cell type (proportion of T cells) multiplied by the rank of that cell type in the innateness gradient.

**Study design.** To study the transcriptome of innate T-cell populations (MAIT, iNKT, Vδ1, and Vδ2), we compared them with adaptive cells (CD4+ T, CD8+ T), as well as NK cells as prototypical innate lymphocytes. Samples used for immunophenotyping and RNA-seq analyses were from healthy individuals. All human sample use was approved by the Brigham and Women's Hospital Institutional Review Board, including written consent for public deposition of RNA sequencing.

For low-input RNA-seq, a matched set of populations were sorted from each individual to avoid batch effects. All blood draws were performed in the morning, and cells were immediately stained and double-sorted directly into lysis buffer. Based on previous RNA-seq analyses on the number of replicates and read depth for optimal differential expression analysis[55], we decided to sort cells from six individuals in duplicate (total of 12 samples per cell type) at a read depth of 4–12 million read pairs (8–24 million reads). The goal of this study was to define the shared transcriptional programs between cell populations rather than variability between individuals. To avoid systematic technical error or batch effects, samples were randomized within the plate for library preparation, and all samples were sequenced together. Five samples were removed for low read depth (described below).

For single-cell RNA-seq, cell populations were sorted from two healthy donors. The same sorting strategy was used for single-cell RNA-seq as was used for low-input RNA-seq, including the use of 5-OP-RU-loaded MR1 tetramers for MAIT cell identification. After sorting, cells were stained with DNA-barcoded hashing antibodies (Biolegend, directed against CD298 and β2-microglobulin) at 0.2 μg/sample, one antibody for each cell type, and a second antibody to barcode each donor. Cells from each donor were pooled in equal proportions, and then analyzed by flow cytometry to ensure viability and representation of each population. Just before loading in the single-cell fluidics device (10X Genomics), cells from the two donors were mixed. Barcodes for cell type and donors are as follows: CD8+ T (GTCAACTCTTTAGCG), CD8+ T (TGATGGCCTATTGGG), MAIT (TTCCGCCTCTCTTTG), iNKT (AGTAAGTTCAGCGTA), Vδ1 (AAGTATCGTTTCGCA), Vδ2 (GGTTGCCAGATGTCA), NK (TGTCTTTCCTGCCAG), Donor 1 (CTCCTCTGCAATTAC), and Donor 2 (CAGTAGTCACGGTCA).

**RNA library preparation and sequencing.** For low-input RNA-seq, Smart-seq2 libraries[56] (poly-A selected) were prepared for the 90 flow-sorted samples (each 1000 cells). These samples were composed of seven main cell types (CD4+ T, CD8+ T, MAIT, iNKT, Vδ1, Vδ2, and NK cells) from six healthy donors, and three additional cell types (δ/αβ, Vδ3, and B cells) from one healthy donor. Each sample had two duplicates. Samples were randomized within the plate. Twenty-five-base paired-end sequencing was performed yielding 4–12 M read pairs (8–24e6 reads, Supplementary Figure 4).

For single-cell RNA-Seq, mRNA and hashing library preparation[57] was performed by the Brigham and Women's Hospital Single Cell Genomics Core using the Chromium Single Cell 3' v2 kit (10x Genomics). The following D7 index primer was used (sample barcode underlined): CAAGCAGAAGACGGCATACGAGAT<u>GACTGACA</u>GTGACTGGAGTTCAGACGTGTGC.

**Low-input RNA-Seq gene expression quantification.** For low-input RNA-seq, we used Kallisto version 0.43.1[58] to quantify gene expression using the Ensembl 83 annotation. We included protein-coding genes, pseudogenes, and lncRNA genes. As expected, protein-coding genes were the most highly expressed, followed by lncRNAs and then pseudogenes (Supplementary Figure 4c). We removed five outlier samples that had a low proportion of common genes detected (one MAIT,

one CD8[+] T, one NK, and two Vδ1 samples; Supplementary Figure 4d). We used log-transformed tpm (transcripts per million) as our main expression measure, which accounts for library size and gene size (specifically log2(tpm + 1)). We considered as expressed genes those with a log2(tpm + 1) > 2 in at least 10 samples. We further performed quantile normalization on the log2(tpm + 1) values for our differential expression analyses. Boxplots were created in R with either boxplot() or ggplot2 geom_boxplot() functions. Unless stated differently in the figure legend, boxes show the first to third quartile with median, whiskers encompass 1.5× the interquartile range, and data beyond that threshold indicated as outliers.

**CITE-seq gene expression quantification and cell hashing**. For CITE-seq, we quantified mRNA and antibody unique molecular identifiers (UMI) counts separately and kept cells that passed QC in both modalities. We quantified gene expression with CellRanger version 2.1.0, mapping reads to the human genome (assembly GRCh38). We removed cells that expressed 500 or fewer unique genes, or had at least 20% of UMIs mapping to mitochondrial genes. We counted cell-hashing antibody UMI counts with CITE-seq count[50], modified to filter out UMIs with fewer than five reads. We removed cells that met any of three antibody exclusion criteria for either the donor or cell-type antibody barcodes: 1, fewer than 10 total antibody UMIs; 2, UMI count for the second-most-abundant antibody greater than 10% of UMI count for the most abundant antibody; 3, UMI count for the most abundant antibody is less than 75% of the total antibody UMI counts. We retained cells that passed the mRNA- and antibody-specific criteria. After filtering, each cell was assigned a cell type and donor based on the most abundant hashing antibody barcode from each set.

Gene expression UMI counts within each cell were normalized for library size (total number of UMIs) and log-transformed (ln(counts per 1e04 + 1)). We performed PCA on the top 1545 most variable genes, ranked by the coefficient of variation, mean centered and scaled by their standard deviation. We performed an L2 normalization to induce cosine distance between cells. Cosine distance has previously been shown to be a more robust distance metric than Euclidean distance for single-cell RNA-seq data[59,60]. We used the function prcomp_irlba()[61] for PCA. For visualization, we used the top 20 principal components to compute the two-dimensional UMAP projection[62]. We ran UMAP with the following parameters: n_neighbors = 30 L, metric = "correlation", and min_dist = 0.1. Cells were clustered based on the top 20 gene expression PCs using shared nearest-neighbor modularity clustering (RunModularityClustering function in Seurat v2[63]). The clustering algorithm was run with a resolution parameter of 0.4 and yielded four clusters.

**Differential expression analyses**. We used linear mixed models or linear models for our differential expression and expression association analyses. For these analyses, we used a likelihood ratio test between two nested models using anova() in R, and a Bonferroni threshold to call significant cases (0.05 divided by the total number of tests).

For low-input RNA-seq, the dependent variable was quantile-normalized log2 (tpm + 1) expression values. Within our predictor variables, we used in all cases donor ID as a random effect. For associations with the innateness gradient, we used one fixed effect composed of integers from 1 to 7 (for CD4[+] T, CD8[+] T, MAIT, NKT, Vδ1, Vδ3, and NK, respectively). In the differential expression between adaptive cells and PLZF[+] ITC, we used one fixed effect taking values of 0 or 1, respectively.

For differential gene expression in single-cell RNA-seq, similar to low-input RNA-seq, we fit a linear model per gene to identify genes associated with the innateness gradient or differentially expressed between cell-type subsets. The dependent variable was log-transformed library-normalized expression values. For associations with the innateness gradient, we used one fixed effect composed of integers from 1 to 7 (for CD4[+] T, CD8[+] T, MAIT, NKT, Vδ1, Vδ3, and NK, respectively), and as covariates number of UMIs per cell (log-transformed), percent of mitochondrial UMIs, and donor (0 or 1 values). Only genes with a nonzero value in at least 100 cells were included. For differential expression between CD8[+] T-cell subsets or Vδ1 subsets, we used one fixed effect taking values 1 for the subset being analyzed or 0 for the other two subsets (for CD8[+] T) or one subset (for Vδ1). We included as covariates the number of UMIs and the percent of mitochondrial UMIs per cell. Only genes with a nonzero value in at least 60 cells were included.

**Gene ontology term enrichment analyses**. We downloaded Ensembl gene IDs linked to Gene Ontology (GO) terms on April 2016[64,65]. This included 9797 GO terms and 15,693 genes. We tested for GO enrichment sorting genes by the β (effect size) of our differential expression analysis. We used the minimal hypergeometric test[66] to test for significance. We confirmed significance of enrichment for the top GO terms using an alternative method: the function gsea() of the liger package (https://github.com/JEFworks/liger).

**Pathway enrichment analysis**. We downloaded genes pertaining to 12 KEGG pathways[67] from the Consensus Pathway Database-human http://cpdb.molgen. mpg.de/[68] in March 2017. First, we calculated the F statistic per expressed gene in our dataset as a metric of variability between cell types. Then we tested whether the F statistics in genes of a certain pathway were higher than the other expressed genes

using a Wilcoxon test. Three pathways had a P-value < 0.05. Since higher expressed genes tend to have higher F statistics, we further tested whether these three pathways had significantly higher F statistics than expected by controlling for gene expression. Specifically, we chose a null set of genes with similar expression levels by taking for each gene in a pathway, 30 random genes with mean level of expression (across all cell types) within 10% of the standard deviation. After this, only the pentose phosphate pathway had genes with F statistics higher than expected (P = 0.018, Wilcoxon test). We further tested enrichment of this pathway in genes associated with innateness gradient using gsea() in the liger package.

**Immunophenotyping associations**. Associations among cell types and clinical traits were performed with Pearson correlation (t test for P-values), and when accounting for different clinical variables (e.g.,, age), were tested with linear regression using cell-type percentages in log scale (t test for P-values). For iNKT cell abundance, there were two individuals with zero values, and these were converted to the next minimal value of 0.01 before log transformation. Covariation between cell types was performed with Spearman rank correlation (t test for P-values).

**PLZF target analysis**. We downloaded PLZF ChIP-seq peaks from the Gene Expression Omnibus (GEO) database from Mao et al.[38] (accession number GSE81772). We used genes from the mouse Gencode vM14 annotation. We defined gene targets as mouse genes with a PLZF peak in the gene body or within 2 kb from the transcription start site (TSS). We downloaded mouse–human gene homologs from BioMart[69]. We selected only genes with 1-to-1 orthologs. We then checked from the mouse PLZF gene targets to which human ortholog they correspond to. Finally, we performed logistic regression to determine whether gene targets are enriched in differentially expressed genes between PLZF[+] ITC and adaptive T cells. Specifically, the response variable is 0 or 1 for nontarget or target gene, respectively. The predictor variable was the β of the differential expression analysis of PLZF[+] ITC versus adaptive T cells. We also tested enrichment defining gene targets if a peak was found only at the promoter region of a gene (−2 kb to + 1 kb from TSS) and found similar results.

**Analysis of public datasets**. From Hertoghs et al.[45], we downloaded their processed microarray expression matrix containing HCMV-specific CD8[+] T-cell samples. If multiple probes were present for a given gene, we calculated the average expression of those probes. From Ranzani et al.[46], we downloaded fastq files for CD4[+] T-cell subsets and processed them as our low-input RNA-seq data, quantifying gene expression with kallisto and using log2(tpm + 1) values. From Fonseka et al.[47], we used processed log2(tpm + 1) expression values from all their CD4[+] T-cell subset samples. From Azizi et al.[49], we used their processed single-cell imputed expression data and their inferred cluster annotations for all T cells and NK-annotated cells (Supplementary Figure 13) or annotated T cells from normal and tumor tissues only (Fig. 7c). For each dataset, an innateness score was calculated per sample or per single cell as described above in the Glossary of innateness terms section of the Methods.

**Flow cytometry and cell sorting**. For immunophenotyping, Ficoll-isolated (GE Healthcare) PBMCs were prepared within 2 h of overnight fasting with blood draw between 8 and 10 AM, stained, and data were acquired the same day. For sorting, freshly isolated PBMCs from donors that had at least 0.1% for each cell type were processed in accordance with the ImmGen standard operating procedure[70]. Briefly, after Fc receptor-binding inhibitor (eBioscience), cells were stained with surface antibodies and dead cells were stained with 7-AAD (Biolegend). Using a FAC-SAria Fusion sorter fitted with a 100 μM nozzle, 1,000 cells double-sorted in duplicate directly V-bottom plates with TCL lysis buffer (Qiagen) and stored frozen until processing. The gating strategy for sorting and validation studies is shown in Supplementary Figure 3.

For validation studies, cryopreserved PBMCs were used from a total of 15 donors. The antibodies used for flow-cytometric validation are listed separately. Data were acquired with a five-laser LSR Fortessa or three-laser FACSCanto II (BD Biosciences) and analyzed with FlowJo (Treestar). A live–dead dye was used for all staining, either 455UV (eBioscience) or ZombieAqua (Biolegend). For intracellular cytokine production studies, cells were fixed with 4% paraformaldehyde, then permeabilized with BD Perm/Wash (BD Biosciences), and stained with intracellular antibodies. For intranuclear staining to assess expression of transcription factors, cells were fixed and permeabilized using the FoxP3 buffer set (eBioscience). For validation studies, MAIT cells were identified as Vα7.2[+]CD161[+] T cells.

**qPCR analysis**. For 47S rRNA quantification, cells were sorted directly into RLT buffer (Qiagen) before RNA extraction (Qiagen, RNeasy). Primers were designed to span the first rRNA-processing site using the following sequences: forward: GTCAGGCGTTCTCGTCTC, reverse: GCACGACGTCACCACAT. HPRT was used as a housekeeping control (forward: CGAGATGTGATGAAGGAGATGG, reverse: TTGATGTAATCCAGCAGGTCAG). qPCR was performed using the Brilliant III Ultra-Fast SYBR QPCR Master Mix (Agilent), read on a Stratagene MX3000P system.

**In vitro cellular studies**. For cellular activation studies, PBMCs were cultured in RPMI 1640 supplemented with 10% FBS (Gemini), HEPES, penicillin/streptomycin, L-glutamine, and 2-mercaptoethanol. Cytokines were from Peprotech except for IFN-α (R&D Systems). For assessment of cytokine production, PMA (200 ng per ml, Sigma) and ionomycin (500 ng per ml, Sigma) were added along with Protein Transport Inhibitor Cocktail (eBioscience) containing brefeldin and monensin for 4 h (Fig. 1) or 2 h (Fig. 4). Actinomycin D (Sigma), cycloheximide (Sigma), or DMSO vehicle (Sigma) were added in some experiments at 1 μM 30 min before activation. IFN-γ transcriptional blockade was confirmed by quantitative PCR. Cytokine production in response to IL-12 (20 ng per ml), IL-18 (50 ng per ml), and IFN-α (50 ng per ml), PBMCs were cultured for 16 h with these cytokines, with eBioscience Protein Transport Inhibitor Cocktail added for the last 4 h of culture. For measurement of cellular ROS, PBMCs were thawed, rested overnight in complete media without added cytokines, followed by the addition of CellRox Green (Thermo Fisher) for 1 h. For proliferation, cells were labeled with CFSE (5 μM for 5 min in PBS), and then cultured at a 2:1 ratio with anti-CD3/CD28-coated beads (Dynabeads, Thermo Fisher). Division index was calculated as (cells divided once + (cells divided twice/2) + (cells divided ≥ 3 times/2.67))/(undivided cells + (cells divided once/2) + (cells divided twice/4) + (cells divided ≥ 3 times/8)) (FlowJo, TreeStar). Migration assays were performed over 3 h using 3 μM Transwell inserts (Corning), with quantification of cell number migrated to the lower chamber.

**Ribopuromycylation**. To assess ribosomal activity, we adapted a microscopic technique, ribopuromycylation[17] for use by flow cytometry. Puromycin was added for 5 min in the presence of emetine (100 μg per ml), followed by fixation with 4% paraformaldehyde, permeabilization with BD Perm/Wash, and staining with an antibody that recognizes puromycin (EMD Millipore).

**List of antibodies**. See Supplementary Data 4.

**Code availability**. The R source code used in key analyses is available at Github.

**Reporting Summary**. Further information on experimental design is available in the Nature Research Reporting Summary linked to this Article.

## Data availability

RNA-seq data that support the findings of this study have been deposited in GEO, with the accession code GSE124731. Data can also be viewed using an interactive browser at https://immunogenomics.io/itc.

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

## Acknowledgements

We thank J. McCluskey, L. Morretta, L. Lynch, the National Institutes of Health Tetramer Facility, and the Kraft Family Blood Donor Center for providing critical materials. We thank members of the Brennan, Raychaudhuri, and Brenner laboratories, as well as S. Suliman, P. Ivanov, S. Lyons, N. Kedersha, and M. Fay for thoughtful discussions and/or critical reading of the paper. We thank S. Pathak for facilitating pilot data. Supported by the National Institutes of Health (AI102945 to P.J.B., AI113046 and AI063428 to M.B.B., U19AI111224, U01GM092691, U01HG009379, and R01AR063759 to S.R.), the Doris Duke Charitable Foundation (2013097 to S.R.), the Violin and Karol families (to P.J.B.), and the Swiss National Science Foundation (Early Postdoc Mobility Fellowship to M.G.-A.).

## Author contributions

M.G.-A., P.J.B., and S.R. conceived, designed, and performed experiments/analysis, wrote the paper, and supervised the research. M.B.B. contributed to experimental design and writing the paper. N.T., A.R.M., H.K., S.H., K.S., G.F.M.W., A.N., R.B.P., and I.K. performed experiments, interpreted data, and contributed to the paper.

## Additional information

**Competing interests:** The authors declare no competing interests.

