## [Peer Review File · Nature Communications]

Reviewers' comments:

Reviewer #1 (Remarks to the Author):

This manuscript describes immune-phenotyping and transcriptional profiling of T cell subsets from peripheral blood from a large number of humans. The authors describe the spectrum of abundance of several understudied cell subsets and relate these to biological variables from the patient cohort. They go on to sort these same cell subsets from a smaller number of volunteers and perform low coverage RNAseq. Subsequent data analysis defines pathway, transcriptional and translational regulation that distinguish these T cell subsets. A novel 'innateness' classifier is developed that is presented as a potential tool to study T cells responses.

The work here is conceptually simple and performed on a large scale. Technical care is evident in the experimental design and the thorough nature of the data analysis.

My principal concern about this manuscript is its entirely descriptive nature. Essentially, there is immunophenotyping with very minimal functional experimentation followed by RNAseq with data analysis. I am unclear as to whether this level of effort and resulting manuscript provide a sufficient advance to warrant publication in a general interest journal such as Nature communications. Perhaps application of the 'innateness' scoring system to a real biological event would convince me otherwise.

More mundane requests:

1. The RNA seq data QC metrics could use some additional information. Specifically I would like to see a supplemental table with the following information: # raw reads per sample, # reads passing quality filter, % of reads mapped, % of reads uniquely mapped, % of duplicate reads. These metrics are necessary for independent interpretation of the data.
2. Although this is a very minor point, I feel the authors are not entirely accurate in their statements about IFN γ production following PMA and IL12/18 treatment (pages 7-8). The authors comment only on % of cells that are positive, while their is undescribed richness in the data in terms of level of IFN γ production. Please discuss in the text.

Reviewer #2 (Remarks to the Author):

In this study, the authors investigated the common features and functional state of innate T cells (ITCs). They found that ITCs possess the heterogeneity, which contain MAIT, iNKT, V δ 1 and V δ 2 4 cell types, and the sum of these cells accounts for mean 8.4% of human T cells. Unbiased transcriptomic analyses suggested that human lymphocytes present the feature of continuous "innateness gradient" process from adaptive cell to innate cells. In terms of biological property, they found that innate cells ITCs present the decreased proliferation potential (ribosome biogenesis, MYC expression) and increased effector function (cytotoxicity, chemokines, Reactive oxygen species metabolism).

This work classifies the human innate T cells and emphasizes important signaling pathways from innate cells to adaptive cells. However, there are some issues to be clarified before consideration of publication

Major concerns:

- (1) What is the developmental relationship among ITCs (MAIT, iNKT, V δ 1 and V δ 2 cells)?
- (2) It would be better to perform single cell RNA-seq using human lymphocytes. sc-RNA analysis may find more novel cell types of ITCs and reveal the heterogeneity and trajectory of pre-defined ITCs.
- (3) The authors claimed that the proliferation of innate T cells is lower than adaptive T cells by

transcriptomic analysis. A direct proliferation analysis should be provided by flow cytometry.

(4) In this study, the PLZF target analysis used the published mouse database, then they evaluated the PLZF expression profiling in MAIT, iNKT, V δ 1 and V δ 2 cells. Because human and mouse are different, it would be better to replenish the PLAF Chip-seq analysis with human T lymphocyte data.

(5) The analysis of translational machinery showed that ribosome biogenesis and proliferation potential are highly associated with adaptiveness, whereas the level of total active translation is higher in innate cells than that in adaptive cells. Therefore it is hard to compare the difference of the translational activity between adaptive cells and innate cells. Further investigation and discussion would be required to verify this point.

Minor concerns:

a. There is a wrong labelling in Fig.2b, the second "V δ 1" should be revised to "V δ 2".

b. In this MS, their results suggest that ITCs express chemokine receptors such as CXCR1, CXCR2, and CCR5 (supplementary Fig.7c). But CXCR1 and CXCR2 are not shown in supplementary Fig.7c.

c. In Fig5, all panels contain the 7 cell types, except for Fig.5f and Fig.5g in which there is no NK cell data. Since NK cells are the representative of innate cells, it could be integrated to make a complete comparison.

Response to reviewer comments for “Lymphocyte innateness is defined by underlying transcriptional states reflecting a balance between proliferation and effector functions” by Gutierrez-Arcelus *et al.*

We would like to thank the reviewers for their thoughtful comments and suggestions. We have carefully considered these comments, and inspired by your suggestions, we generated new transcriptional and functional data alongside additional analyses of public data sets. We find this new data exciting, and we believe that it augments the manuscript substantially. We genuinely appreciate your suggestions which we believe have significantly strengthened our manuscript.

Individual reviewer comments are addressed individually following a summary of the additions and changes to our manuscript:

1. Single-cell RNA-seq of CD4⁺ T, CD8⁺ T, iNKT, MAIT, V δ 1, V δ 2, and NK cell populations. This new data recapitulated the innateness gradient that we had identified with low-input RNA-seq, and additionally revealed four ‘innate states’ separated by differential expression of innateness gradient genes. Most canonical populations showed heterogeneity and variably populated one or more state of innateness.

- **Figure 8**

- **Supplementary Figures 14-17**

2. Application of the gene signature derived from our innateness gradient to human disease. We analyzed public human expression profiling datasets, including bulk and single cell transcriptional data from healthy donors, HCMV-infection, rheumatoid arthritis, and breast cancer. In each case, the innateness signature was evident in the data. Our analyses suggest that the innateness signature is a valuable metric that may have important implications for prognosis in human disease.

- **Figure 7**

- **Supplementary Figures 12 and 13**

3. New functional studies. We performed additional functional characterization and testing of innate T cells to validate findings in our dataset. An important concept that emerged from our data was the trade-off between proliferation and effector function, and the suggestion that this was transcriptionally ‘pre-loaded.’ On the first submission, we showed that proliferation was impaired in innate T cells. We have now further functionally explored the flip-side of this coin, enhanced effector functions. We now demonstrate experimentally that early IFN- γ production is largely the result of pre-formed mRNA. In essence, adaptive cells prioritize use of their translational machinery to make more ribosomes for cell division, while innate cells have pre-formed effector mRNAs to provide rapid effector functions. In addition, we have now performed migration assays to experimentally test the prediction that innate T cells can migrate to sites of inflammation alongside myeloid cells such as monocytes.

- **Figure 4c,d** (pre-formed *IFNG* mRNA)

- **Supplementary Figure 10** (migration)

4. Revision of the manuscript text, including the abstract, results section, and discussion. The text has been substantially revised to highlight the new transcriptional and functional findings of our work, as well as to clarify points raised by reviewers.

Response to reviewer #1: *(Reviewer comments in italics)*

This manuscript describes immune-phenotyping and transcriptional profiling of T cell subsets from peripheral blood from a large number of humans. The authors describe the spectrum of abundance of several understudied cell subsets and relate these to biological variables from the patient cohort. They go on to sort these same cell subsets from a smaller number of volunteers and perform low coverage RNAseq. Subsequent data analysis defines pathway, transcriptional and translational regulation that distinguish these T cell subsets. A novel 'innateness' classifier is developed that is presented as a potential tool to study T cells responses.

The work here is conceptually simple and performed on a large scale. Technical care is evident in the experimental design and the thorough nature of the data analysis.

My principal concern about this manuscript is its entirely descriptive nature. Essentially, there is immunophenotyping with very minimal functional experimentation followed by RNAseq with data analysis. I am unclear as to whether this level of effort and resulting manuscript provide a sufficient advance to warrant publication in a general interest journal such as Nature communications. Perhaps application of the 'innateness' scoring system to a real biological event would convince me otherwise.

We thank the reviewer for their comments, as they provided an important perspective that led us to perform multiple additional experiments and to revise the text and emphasis of the manuscript. We have expanded upon the functional evidence and demonstrated how the innateness score might ultimately lead to disease insight. In particular: 1) We have performed new functional experiments showing the importance of pre-formed mRNA in rapid IFN- γ production. 2) We have performed migration assays to functionally demonstrate that innate T cells migrate towards inflammation through the same chemokine receptors as found on myeloid cells. 3) We have analyzed multiple single-cell and bulk RNA-seq datasets to test in silico how the innateness gene signature that we identified can inform us on disease state and survival.

1) Functional experiments for cytokine production show a role for pre-formed mRNA.

A central global finding of our work is the preferential expression of effector protein-encoding mRNAs in innate T cells, in contrast to rRNA and translational machinery upregulation in adaptive T cells. Notably, we could not detect resting innate T cells actively making many effector proteins for which mRNA was present, for example, IFN- γ and other cytokines. Similarly, although adaptive cells are transcriptionally poised for division, the vast majority of these cells are not actively dividing. We showed in our initial submission that adaptive cells proliferate more rapidly to TCR activation than innate cells. We hypothesized that both innate and adaptive cells are producing RNA for their 'anticipated' response to activation. To test

whether this baseline, pre-formed mRNA contributes to effector responses, we treated primary human cells with either actinomycin D, which blocks transcription, or cyclohexamide, which blocks translation. As expected, since we could not detect IFN- γ protein at baseline, cyclohexamide blocked new cytokine production upon stimulation. Blocking new transcription with actinomycin D, on the other hand, only partially reduced rapid IFN- γ production, consistent with an important role for pre-formed mRNA. These experiments are shown in **Figure 4**. We have added the following text to describe these results:

Although we could detect mRNA for cytokines, intracellular staining did not reveal baseline cytokine protein production in ITC or adaptive T cell populations. Pre-formed mRNA for IFN- γ and/or granzyme B have been demonstrated in NK cells, iNKT cells, and CD8⁺ memory T cells in mice²⁷⁻²⁹. We hypothesized that pre-formed mRNAs might contribute to rapid cytokine production across ITCs. To test this hypothesis, we activated PBMCs with PMA and ionomycin for 2 hrs in the presence of actinomycin D, which blocks new transcription, or cycloheximide, which blocks translation. Early IFN- γ production by all lymphocyte subsets was nearly completely blocked by cyclohexamide, but only partially blocked by actinomycin D (**Fig 4c,d**). These experiments show that pre-formed mRNA contributes to early cytokine production by ITCs, adaptive T cells, and NK cells. Our data suggest that translation of pre-formed effector mRNA may be one of the mechanisms that enable the characteristic rapid response of ITC populations.

2) Migration assays show ITCs migrate towards 'myeloid' chemokines.

Several of the gene sets identified as associated with innateness identified migration as enriched, and in particular, myeloid cell migration. This makes sense for an effector cell population poised for effector responses, and part of their role in host defense likely includes trafficking to peripheral sites of inflammation. Adaptive cells, on the other hand, showed higher expression of CCR7, governing lymphoid recirculation, consistent with their role in lymph nodes. To test the functional capacity of innate T cells to migrate towards classical myeloid chemokines, we performed transwell migration assays to CCL2, CCL3, CCL4, and CCL8. As predicted by the chemokine receptor expression patterns identified in our transcriptional studies, PLZF⁺ innate T cells migrated to these chemokines, while migration of adaptive cells was largely limited to CCL19 (CCR7 ligand). We have added **Supplementary Figure 10** and following text to describe these results:

PLZF expression in T cells was also associated with the aggregate expression of all cytokine and chemokine receptor activity genes (**Supplementary Fig. 9b**), and we validated the expression of several of these receptors by flow cytometry (**Supplementary Fig. 9c**). We tested migration across a permeable membrane to a panel of chemokines, including CCL19 which directs lymphoid recirculation by CCR7 and CCL2, 3, 4, and 8 targeting chemokine receptors shared between ITCs (**Supplementary Fig. 9c**) and myeloid cell populations such as monocytes. PLZF⁺ ITCs readily migrated in response to classical monocyte chemokines, and also in response to CCL19, while migration of adaptive T cells and the PLZF⁻ V δ 1 population was limited to CCL19 (**Supplementary Fig. 10**).

3) Application of the innateness gene signature in human disease.

We thank the reviewer for the interesting suggestion to look in human disease. Another important theme in our paper is that 'innateness' is also a feature of some adaptive T cell subsets. This is particularly evident in our new single-cell RNA-seq data, described further

below in the response to reviewer #2. In our initial submission, we had included analysis of published datasets for HCMV-reactive T cell subsets, and also a dataset with healthy donor CD4⁺ T cells. We have now expanded this area substantially to look at CD4⁺ T cells from a low-input RNA-seq dataset from peripheral blood of patients with rheumatoid arthritis and osteoarthritis from our group (Fonseka et al, *Science Translational Medicine* 2018). We calculated an “innateness score” per sample, based on our PCA gene loadings. As observed in previous datasets, naïve CD4⁺ T cells had the lower innateness score, followed by regulatory T, central memory T, and effector memory T cells (**Figure 7b**). We found that within the CD4⁺ T effector subsets, the one with highest innateness score is the one that is expanded in rheumatoid arthritis patients compared to controls, which we believe to be playing a key role in pathogenesis.

We next analyzed single-cell RNA-seq public dataset derived from breast cancer patients, collecting a total of 45,000 immune cells coming from tumor, normal breast tissue, blood, and lymph node (Azizi et al, *Cell* 2018). We have applied the innateness score to each cell. As shown in **Supplementary Fig. 13**, we recapitulated the innateness gradient in the T cell and NK annotated clusters. Importantly, we have also shown that tumor-derived T cells have a higher innateness on a per-cell basis than cells from normal breast tissue (**Figure 7c**).

We have added / expanded **Figure 7**, added **Supplementary Figure 13**, and the following text:

We also analyzed published RNA-seq data for CD4⁺ T cell subsets from healthy individuals⁶⁴. CD4⁺ effector memory T cells had a higher innateness score than CD4⁺ central memory T cells ($P = 0.008$), which had a higher innateness score than CD4⁺ naïve T cells ($P = 0.032$, **Supplementary Fig. 12c-e**). To investigate innateness in CD4⁺ T cells during inflammation, we calculated the innateness score for CD4⁺ T cell populations profiled by low-input RNA-seq from patients with arthritis⁶⁵, which is thought to be driven in part by CD4⁺ T cells⁶⁶. The innateness score ordered T cell populations from naïve to central memory to effector cells (**Fig. 7b**). The subset that scored highest in innateness, CD4⁺HLA-DR⁺CD27⁻ T cells, are the precise subset that is most expanded in rheumatoid arthritis and correlate with treatment response⁶⁵.

We next interrogated lymphocytes from a single-cell RNA-seq dataset from patients with breast carcinoma⁶⁷. As seen for HCMV-specific CD8⁺ T cells (**Fig. 7a**) and CD4⁺ T cells from rheumatoid arthritis (**Fig. 7b**), lymphocyte populations from patients with breast carcinoma, as annotated in the original study, recapitulated the innateness gradient (**Supplementary Fig. 13**). Further, T cells from within the tumor showed a higher innateness score than those in healthy tissue ($P = 2e-66$), suggesting that T cell innateness was enriched at the site of disease (**Fig. 7c**).

We revised the discussion regarding how the innateness score might allow us to think about immune cells in a disease context:

Finally, the innateness gradient reported here could be applied in different scenarios in order to better understand human immunology and human disease. A transcriptomic innateness score could be employed as a unified T cell metric to classify individual single cells assayed with single-cell RNA-seq, providing a better understanding of patient heterogeneity. We provide data that innateness in the T cell compartment increases with infection, inflammation, and in cancer. The ‘innateness score’ (**Fig. 7**) may be a useful prognostic indicator, and understanding the molecular mechanisms associated with innateness may pave the way for novel therapies aimed at modulating this coordinated transcriptional response.

More mundane requests:

1. The RNA seq data QC metrics could use some additional information. Specifically I would like to see a supplemental table with the following information: # raw reads per sample, # reads passing quality filter, % of reads mapped, % of reads uniquely mapped, % of duplicate reads. These metrics are necessary for independent interpretation of the data.

We have added **Supplementary Table 3** specifying the number of sequenced reads that passed filter, percent mapped reads, percent uniquely mapped reads, percent duplication, plate position, and whether the sample passed our QC or not. The sequencing data we obtained was for reads that already passed the Illumina filter, so we were not able to provide the specific number of reads before filtering for each sample. However, we know that per sequenced lane, 85-86% of reads passed the filter. And within the filtered reads, 92.3% of bases have a base quality score of $\geq Q30$.

Although this is a very minor point, I feel the authors are not entirely accurate in their statements about IFN γ production following PMA and IL12/18 treatment (pages 7-8). The authors comment only on % of cells that are positive, while their is undescribed richness in the data in terms of level of IFN γ production. Please discuss in the text.

Thanks for this suggestion. Indeed, important additional information was uncovered with the suggested analysis. Specifically, a subset of CD8⁺ T cells showed higher IFN- γ MFI compared with some innate T cell subsets in response to strong activation with PMA and ionomycin, which is important and interesting. In our new single-cell RNA-Seq data, some CD8⁺ T cells clustered closer to NK cells than did iNKT cells and MAIT cells. For cytokine-driven activation, the IFN- γ MFI data looked similar to the percent-positive data presented in the initial submission. We have added plots showing IFN- γ MFI in **Supplementary Figure 2** and have added these results to the manuscript.

Response to reviewer #2:

In this study, the authors investigated the common features and functional state of innate T cells (ITCs). They found that ITCs possess the heterogeneity, which contain MAIT, iNKT, V δ 1 and V δ 2 4 cell types, and the sum of these cells accounts for mean 8.4% of human T cells. Unbiased transcriptomic analyses suggested that human lymphocytes present the feature of continuous “innateness gradient” process from adaptive cell to innate cells. In terms of biological property, they found that innate cells ITCs present the decreased proliferation potential (ribosome biogenesis, MYC expression) and increased effector function (cytotoxicity, chemokines, Reactive oxygen species metabolism).

This work classifies the human innate T cells and emphasizes important signaling pathways from innate cells to adaptive cells. However, there are some issues to be clarified before consideration of publication

Major concerns:

(1) What is the developmental relationship among ITCs (MAIT, iNKT, V δ 1 and V δ 2 cells)?

MAIT and iNKT cells are thought to derive from the same thymic progenitors as adaptive $\alpha\beta$ T cells. Coming from the same thymic T cell pool, they are activated through their TCRs by poorly-defined self-ligands presented by their restriction elements (CD1d for iNKT cells, MR1 for MAIT cells), after which they undergo expansion and differentiation (Godfrey, *Nat Immunol* 2010; Gapin, *J Immunol* 2014; Koay, *Immunol Cell Biol* 2018). How their thymic differentiation occurs is of intense interest and this is the subject of ongoing studies in mice. For human $\gamma\delta$ T cells, thymic development remains poorly understood and may depend on both TCR and environmental stimuli (Fahl, *J Immunol* 2014; Wilcox, *Front Immunol* 2018). We are not aware of any developmental lineage that separates human ITCs from adaptive T cells. Rather, the literature suggests that these cell types develop independently from the same thymic precursors as adaptive T cells in response to signals received during development. Whether the developmental signals received by ITCs share a common signaling pathway, and how early these cells acquire their innate phenotype are fascinating questions that warrant future study. We have commented on the developmental relationship in the introduction section:

ITCs develop from the same thymic progenitor cells as adaptive T cells, and each of these populations is thought to develop independently.

(2) It would be better to perform single cell RNA-seq using human lymphocytes. sc-RNA analysis may find more novel cell types of ITCs and reveal the heterogeneity and trajectory of pre-defined ITCs.

We thank the reviewer for the excellent suggestion of using single cell RNA-seq from human lymphocytes. We are convinced of the power of single-cell studies, and as you can probably imagine, we have wanted to perform these experiments for some time. Your suggestion provided timely encouragement. Importantly, we believe that this additional data has strengthened the paper, as described below.

We have sorted a complete set of 7 cell types included in the study from two individuals and performed single cell RNA-seq with the CITE-seq protocol and 10X platform, using a DNA-barcoded antibody cell hashing method to label cell types and individuals. In this way, we could ensure equal representation and confident identification of each cell type. The hashing protocol allowed the cells to be processed in a single 10x run and library preparation, and in the process completely obviate potential batch effects. As with the low-input RNA-seq experiments, the innateness gradient was a primary feature of the data that was evident along PC1 of the PCA performed on the most variable genes (**Supplementary Figure 16**). As with the low-input RNA-seq data, innate T cells in single-cell experiments segregated between NK cells and adaptive T cells. Single-cell data thus provided a compelling validation of the low-input RNA-seq data, showing that the signals we observed were driven by the majority of the population and not a

few outlier cells. For example, the ribosomal gene signature in adaptive cells was seen in virtually the entire adaptive T cell population, not a small percentage of proliferating cells (**Supplementary Figure 16**).

Dimensionality reduction and clustering (**Figure 8**) showed 4 clusters that we propose represent 'states of innateness.' Briefly summarized, the major populations in these 4 states are 1) adaptive T, 2) MAIT and iNKT, 3) $\gamma\delta$ T, and 4) NK cells. However, our single cell data also provided evidence for heterogeneity in the innateness within single populations. For most of the cell types, individual cells populated more than one state of innateness. For example, while CD8⁺ T cells were most abundant in the adaptive cluster, sizeable populations were also found in two more-innate clusters, interspersed with iNKT/MAIT cells or $\gamma\delta$ T cells. The new single-cell RNA-seq data is presented in **Figure 8** and supplementary **Figures. 14-17**.

We will publicly share the low-input RNA-seq dataset and the single-cell dataset with the field. There are many directions that can be pursued with this data, and we are enthusiastic that our datasets will drive important new discoveries by multiple investigators. We have added a browsable feature to our ITC website where users can visualize gene expression in single cells on UMAP space and will make this browsable resource public upon acceptance of this manuscript:

<https://immunogenomics.io/itc/>

user: tcell

password: gradient

We added the following text to describe these results to the main text of the manuscript.

To assess heterogeneity among the lymphocyte populations studied, we applied single-cell RNA-seq to the same 7 cell populations studied above. We first sorted CD4⁺ T, CD8⁺ T, iNKT, MAIT, V δ 1, V δ 2, and NK cells from two donors and then labeled each population with DNA-barcoded antibodies, allowing us to identify the original cell populations using a cell 'hashing' protocol⁶⁸. We then assayed all populations together in a single pooled droplet-based assay, which obviated any batch effects that might have occurred if the populations were assayed separately⁶⁹. After stringent quality control, we obtained 2,036 high-quality single-cell libraries with similar representation of each starting cell population (**Supplementary Table 5, Supplementary Fig. 14**).

Before exploring heterogeneity, we wanted to ensure that the broad trends seen in the bulk RNA-seq data were replicated in the single cell RNA-seq data. We first compared the single-cell and low-input RNA-seq datasets on a gene-by-gene basis. From our single-cell data, we calculated the level of innateness (β based on the association with innateness rank order determined in low-input RNA-seq) for the significant adaptiveness and innateness genes identified in the low-input RNA-seq data. Unsurprisingly, the single-cell gene expression associations with the innateness gradient replicated our low-input RNA-seq data (**Fig. 8a** Spearman rho = 0.82, 94% concordance in direction of effects). Key innateness genes such as *GZMB*, *PRF1*, *KLRD1*, *NKG7*, *HOPX*, *ZEB2*, *ID2*, and adaptiveness genes such as *RPL36*, *RPL22*, *RPS19*, *EEF3E*, *MYC* were also significant in single-cell expression data (all $P < 7e-18$).

For dimensionality reduction and data visualization we applied PCA to the single cell data using the top 1,000 variable genes. PCA of single-cell RNA-seq data (sc-PC1 and sc-PC2) mirrored those obtained for low-input RNA-seq, with sc-PC1 separating adaptive cells from innate cells. Among the innate populations, NK cells were on one end, iNKT and MAIT cells on the other end, and $\gamma\delta$ T cells in between (**Supplementary Fig. 15a**). Unsurprisingly, PC1 from the single-cell data (sc-PC1, **Supplementary Fig. 15b**) largely recapitulated the innateness gradient

derived from low-input RNA-seq (**Fig. 2b**), with the position of V δ 1 and V δ 2 reversed. The innateness order for V δ 1 and V δ 2 was different between the two donors included in the single-cell study, suggesting possible donor to donor variability among $\gamma\delta$ T cells (**Supplementary Fig. 15c**). The innateness score, calculated for each single cell using PC1 loadings from the low-input bulk RNA-seq data was highly correlated with sc-PC1 ($r = 0.9$, **Supplementary Fig. 15d**).

To better understand cellular heterogeneity, we embedded and clustered single cell data. We embedded single cell data, using the top 20 principal components, into a two-dimensional space with uniform manifold approximation and projection (UMAP). Adaptive T cells to NK cells were spread across the UMAP (**Fig. 8b**). Key innateness and adaptiveness genes colored a gradient across the UMAP (**Supplementary Fig. 16**), as did the innateness score calculated per cell (**Fig. 8c**). To define underlying innateness states at the single cell level, we applied shared nearest neighbor modularity clustering which identified 4 distinct clusters. We propose that these groups represent 4 'states of innateness' that can be adopted by lymphocytes. The most adaptive of the four clusters, Cluster 1 contained mostly CD4⁺ T and CD8⁺ T cells. Cluster 2 contained mainly iNKT, MAIT, and V δ 2 T cells. Cluster 3, was even more innate, and contained predominantly V δ 1 and V δ 2 T cells. Finally, Cluster 4 was completely innate, consisting mostly of NK cells (**Fig. 8d**).

For some cell types, we noted that sizeable subsets were present in multiple clusters (**Fig. 8e, Supplementary Fig. 17**). CD8⁺ T cells exhibited the most striking heterogeneity, showing populations that clustered into three different states: CD8-1 clustered with adaptive CD4⁺ T cells, CD8-2 which was most similar to iNKT and MAIT cells, and CD8-3 cells which were most similar to $\gamma\delta$ T cells (**Fig. 8f**). When we examined the genes that separated each CD8⁺ T cell cluster, we identified genes that were part of the innateness and adaptiveness genes identified in our low-input RNA-seq data (182 out of 237 unique genes that were significant in any of the three one-vs-all differential expression comparisons, $P < 4.4e-05$). The CD8-2 population was characterized by intermediate expression of innateness and adaptiveness genes, rather than by expression of unique genes (**Fig. 8g**). Only 4 genes, GZMK, TMSB10, CCL5, and GMC1 were significantly differentially expressed in the CD8-2 population. V δ 1 T cells were predominantly in Cluster 3 with V δ 2 T cells, although a substantial V δ 1 population also populated Cluster 1 with adaptive CD4⁺ T cells (**Supplementary Fig. 17**). 94% of genes that distinguished this adaptive-like V δ 1 population ($P < 5.2e-05$) were adaptiveness and innateness genes, including downregulated cytotoxic genes such as *PRF1* and *KLRD1*, and upregulated ribosomal machinery genes such as *RPL34*, *RPL22* and *EEF1A1*. This finding is consistent with recent reports on V δ 1 populations being highly variable across donors, and with dynamic changes in response to infection, suggesting that a naive V δ 1 state may be part of their development^{70,71}. Besides NK cells, and MAIT cells, most of the lymphocytes assayed spanned multiple clusters. Taken together, our single-cell RNA-seq data demonstrates that the innateness cell-type hierarchy seen in low-input RNA-seq reflects an average of multiple cell states, and that individual innate and adaptive T populations variably populate these innateness states.

(3) The authors claimed that the proliferation of innate T cells is lower than adaptive T cells by transcriptomic analysis. A direct proliferation analysis should be provided by flow cytometry.

We apologize that this experimental data was not clearly explained in the results section. In **Figure 5f,g**, we have performed a proliferation assay with anti-CD3/CD28 beads on primary cells labeled with a dye that equally partitions to divided daughter cells (CFSE), which allows for direct measurement of proliferation. We have shown that the proliferation index follows the gradient with CD4⁺ and CD8⁺ T cells proliferating faster than ITCs. To clarify, we have edited the main text, and have included the following description:

Since new ribosome production is necessary for proliferation, and *MYC* expression is generally associated with proliferative capacity, we hypothesized that proliferation potential might associate with adaptiveness. We assayed proliferation in primary human T cells in response to anti-CD3/CD28-coated beads, by measuring (carboxyfluorescein succinimidyl ester) CFSE dye dilution. NK cells were omitted from this analysis since they do not respond to anti-CD3/CD28-coated beads. Like *MYC* and ribosome biogenesis, proliferation was associated with

adaptiveness and innate T cells proliferated less than adaptive T cells (Spearman rho = -0.73, P = 5.8e-04, **Fig. 5f,g**). These results recall the well-described regulation of ribosomes in prokaryotes, where ribosome biogenesis is major energetic control point, is suppressed in conditions under which growth and division are deprioritized³², and can be fine-tuned to ensure maximal occupancy of active ribosomes³³.

(4) In this study, the PLZF target analysis used the published mouse database, then they evaluated the PLZF expression profiling in MAIT, iNKT, Vδ1 and Vδ2 cells. Because human and mouse are different, it would be better to replenish the PLZF Chip-seq analysis with human T lymphocyte data.

We agree with the reviewer that it would be both interesting and valuable to the field to have PLZF target analysis on individual subsets of human cells. We point out in the manuscript that the expression of reported PLZF-induced genes differs between PLZF⁺ T cells and NK cells (also PLZF⁺). The expression of these genes must then depend on factors in addition to PLZF. To address this point in humans would require a very demanding study, with ChIP-seq on multiple cell populations, some of which are low-abundance, and from multiple donors. Although these are certainly interesting experiments, we believe that they are beyond the scope of this report, and chose to focus our resources on single-cell RNA-seq and additional functional analysis.

(5) The analysis of translational machinery showed that ribosome biogenesis and proliferation potential are highly associated with adaptiveness, whereas the level of total active translation is higher in innate cells than that in adaptive cells. Therefore it is hard to compare the difference of the translational activity between adaptive cells and innate cells. Further investigation and discussion would be required to verify this point.

This is an important point concerning one of the central findings of this paper. When we initially observed lower expression of ribosomal and translational components in innate T cells, one possibility that we considered was that these cells were maintained in a quiescent state. The ribopuromylation experiment disproved this hypothesis, as the innate T cells were more translationally active overall. The difference in ribosomal protein-encoding mRNAs and rRNA reflects what type of transcripts each cell type is prioritizing. Ribosome biogenesis can likely be attributed to MYC, known to direct ribosome production. Further, we have added new transcriptional data showing that RNA polymerase I, responsible for rRNA production, is expressed at higher levels in adaptive cells, while RNA polymerase II is expressed at higher levels in innate cells (**Supplementary Figure 8**).

We further explored the role of baseline effector mRNA production in effector cells by isolating the contribution of pre-formed *IFNG* mRNA to rapid IFN-γ protein production. As mentioned above in response to reviewer #1, we treated primary human cells with either actinomycin D, which blocks transcription, or cyclohexamide, which blocks translation. As expected, since we could not detect IFN-γ protein at baseline, blocking translation with cyclohexamide blocked new cytokine production after stimulation. Blocking new transcription with actinomycin D, on the

other hand, only partially reduced rapid IFN- γ production, consistent with an important role for pre-formed mRNA. These experiments are shown in **Figure 4**.

Overall, these findings provide new evidence that one of the mechanisms by which ITCs ensure a rapid response is by increasing the amount of pre-formed mRNA for effector function genes that will be promptly translated upon activation. In this context, having a higher amount of active translation, as shown by the ribopuromycylation assay, could further enhance this rapid-responder potential.

We propose a model in which innate lymphocytes are pre-loaded with mRNA for rapid effector function, while adaptive cells are pre-loaded with ribosomal protein-encoding mRNA and rRNA needed to produce ribosome for proliferation. We have also revised and expanded this topic in the discussion section with the following text:

Our identification that mRNAs encoding for ribosome subunits and other factors involved in translation associated with adaptiveness (**Fig. 5**) also sheds light on what it means to be innate. In a given cell, loss of rapid proliferative capacity may be an inherent trade-off for enhanced effector function, a balance previously observed for CD8⁺ T cells²⁸. For an adaptive T cell, population expansion is of central importance during immune responses. ITCs, on the other hand, are likely to function as sentinels early during infection, acting to enhance ensuing immune responses in response to microbial molecules. For such a role, rapid effector responses are key, and proliferation may serve only to replenish numbers at a later stage. Taken together, the effector-focused transcriptional programs of ITCs and proliferation-focused programs of adaptive cells are ideally suited to support their respective roles in immunity. It is notable that effector gene programs in ITCs and proliferative gene programs in adaptive cells were present at rest. Our data showing that a substantial portion of early IFN- γ is produced from pre-formed mRNA (**Fig. 4c-d**), together with higher overall active translation in ITCs (**Fig. 5i**), supports the concept of ITCs being in a poised state ready for a rapid and efficient effector response. Similarly, we postulate that adaptive cells are poised for division, with relatively higher RNA polymerase I, *MYC* expression, and prioritization of ribosome formation that would be needed for cell proliferation.

Minor concerns:

a. There is a wrong labelling in Fig.2b, the second "V δ 1" should be revised to "V δ 2".

Thanks very much for pointing this out. We have corrected the mislabeling.

b. In this MS, their results suggest that ITCs express chemokine receptors such as CXCR1, CXCR2, and CCR5 (supplementary Fig.7c). But CXCR1 and CXCR2 are not shown in supplementary Fig.7c.

Although CXCR1 and CXCR2 are also significantly associated with the innateness gradient, we meant to indicate CCR1 and CCR2 in the manuscript, for which we had performed flow cytometric validation. We have corrected this mistake in the manuscript. For this revision, we also performed migration assays with chemokines that activate through CCR1 and CCR2 (**Supplementary Figure 10**).

c. In Fig5, all panels contain the 7 cell types, except for Fig.5f and Fig.5g in which there is no NK cell data. Since NK cells are the representative of innate cells, it could be integrated to make a complete comparison.

For these functional proliferation assays, we used anti-CD3 and anti-CD28 beads to activate T cells. NK cells do not respond to this method of activation. We are not aware of a 'fair' way to compare the proliferation of NK cells and T cells, so we chose to omit this cell type from proliferation analyses. We have clarified this in the results section (please see text quoted in response to point (3) above).

REVIEWERS' COMMENTS:

Reviewer #1 (Remarks to the Author):

I thank the authors for the response to suggestions. The revised manuscript is substantially improved.

The inclusion of QC metrics for the sequencing data adds to reader's ability to interpret the data. the new functional experiments and extension to pathology is a nice addition.

Paul Wade

Reviewer #2 (Remarks to the Author):

I am satisfied with the revision in that most of my concerns have been well addressed.

Response to reviewer comments:

Regarding our revised manuscript entitled “Lymphocyte innateness defined by transcriptional states reflects a balance between proliferation and effector functions” by Gutierrez-Arcelus *et al.* (NCOMMS-18-09292A), the reviewers did not have any new suggestions or recommendations, so no reviewer-directed changes were made.